# Natural gradient enables fast sampling in spiking neural networks

**Paul Masset**[1,2]*, **Jacob A. Zavatone-Veth**[1,3]*, **J. Patrick Connor**[4],
**Venkatesh N. Murthy**[1,2]†, **Cengiz Pehlevan**[1,4]†

[1]Center for Brain Science, [2]Department of Molecular and Cellular Biology,
[3]Department of Physics, [4]John A. Paulson School of Engineering and Applied Sciences,
Harvard University
Cambridge, MA 02138
`paul_masset@fas.harvard.edu, jzavatoneveth@g.harvard.edu,`
`cpehlevan@seas.harvard.edu`

## Abstract

For animals to navigate an uncertain world, their brains need to estimate uncertainty at the timescales of sensations and actions. Sampling-based algorithms afford a theoretically-grounded framework for probabilistic inference in neural circuits, but it remains unknown how one can implement fast sampling algorithms in biologically-plausible spiking networks. Here, we propose to leverage the population geometry, controlled by the neural code and the neural dynamics, to implement fast samplers in spiking neural networks. We first show that two classes of spiking samplers—efficient balanced spiking networks that simulate Langevin sampling, and networks with probabilistic spike rules that implement Metropolis-Hastings sampling—can be unified within a common framework. We then show that careful choice of population geometry, corresponding to the natural space of parameters, enables rapid inference of parameters drawn from strongly-correlated high-dimensional distributions in both networks. Our results suggest design principles for algorithms for sampling-based probabilistic inference in spiking neural networks, yielding potential inspiration for neuromorphic computing and testable predictions for neurobiology.

## 1 Introduction

Neural circuits perform probabilistic computations at the sensory, motor and cognitive levels [1–4]. From abstract representations of decision confidence [5] to estimates of sensory uncertainty in visual cortex [6, 7], evidence of probabilistic representations can be found at all levels of the cortical processing hierarchy [8]. To be behaviorally useful, these probabilistic computations must occur at the speed of perception [9]. However, how neuronal dynamics allow brain circuits to represent uncertainty in high-dimensional spaces at perceptual timescales remains unknown [3, 10–12].

Several neural architectures for probabilistic computation have been proposed, including: probabilistic population codes [13], which in certain cases allow a direct readout of uncertainty; direct encoding of metacognitive variables, such as decision confidence [4, 5, 8]; doubly distributional codes [14, 15] which distinguish uncertainty from multiplicity; and sampling-based codes [9, 16–25], where the variability in neural dynamics corresponds to a signature of exploration of the posterior probability. Most experiments quantifying uncertainty representations in single biological neurons have only

---

*PM and JAZ-V contributed equally to this work.
†VNM and CP jointly supervised this work.

36th Conference on Neural Information Processing Systems (NeurIPS 2022).

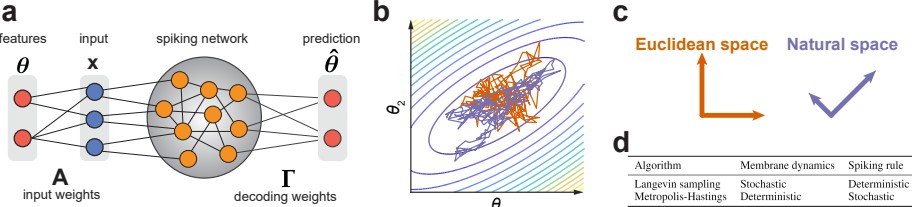

Figure 1: Probabilistic inference in spiking neural networks. **a**. Circuit diagram. **b**. Langevin sampling in Euclidean space and natural space. **c**. Geometry of the inference space in (b). Arrows indicate the principal directions of the sampling noise covariance. **d**. Comparison of Langevin and Metropolis-Hastings sampling algorithms.

varied parameters along one or two dimensions, such as in Bayesian cue combination [2, 10, 26]. In these conditions, many algorithms can perform adequately. However, probabilistic inference becomes more challenging as the entropy of the posterior distribution—which often scales with dimensionality—increases [27, 28]. Some algorithms that work well in low dimensions, such as probabilistic population codes, may scale poorly to high-dimensional settings [3].

Of the proposed approaches to probabilistic computation in neural networks, sampling-based codes are grounded in the strongest theoretical framework [27, 29–33], and have been used to perform inference at scale [34–36]. Moreover, they predict specific properties of neural responses in visual cortex, including changes in Fano factor and frequency of oscillations with tuning and stimulus intensity [23]. Previous works have proposed several approaches to accelerate sampling in biologically-inspired algorithms. Hennequin et al. [9] showed that adding non-reversible dynamics to rate networks can reduce the sample autocorrelation time. However, they did not study convergence of the sampling distribution, and did not consider the biologically-relevant setting of spiking networks. Savin and Denève [20] used a distributed code to parallelize sampling in spiking networks, but only considered two-dimensional distributions. Therefore, it remains unclear how accurate sampling from high-dimensional distributions at behaviorally-relevant timescales can be achieved using spiking networks.

In this paper, we show how the choice of the geometry of neural representations at the population level [37, 38], set by the neural code and the neural dynamics, can accelerate sampling-based inference in spiking neural networks. Ideas from information geometry allow us to perform inference in the natural space of parameters, which is a manifold with distances measured by the Fisher-Rao metric (Figure 1.c) [39–42]. Concretely, we leverage recently-proposed methods for accelerating sampling from the machine learning literature [28, 42–44] to design novel efficient samplers in spiking neural networks. The structure and major contributions of this paper are divided as follows:

- In §2, we construct from first principles a novel spiking neural network model for sampling from multivariate Gaussian distributions. This model is based on a probabilistic spike rule that implements approximate Metropolis-Hastings sampling. We show that efficient balanced networks (EBNs) [20, 45–49] emerges as a limit of this model in which spiking becomes deterministic.

- In §3, we show that population geometry enables rapid sampling in spiking networks. Leveraging the "complete recipe" for stochastic gradient MCMC [43], we establish principles for the design of efficient samplers in spiking neural networks. Then, we show how neural population geometry enables fast sampling—on the timescale of tens of milliseconds—in two limits of the model introduced in §2: EBNs in which sampling is driven by stochastic Langevin dynamics [20], and networks in which sampling is driven purely by a Metropolis-Hastings probabilistic spiking rule.

- Finally, in §4 we conclude by discussing the implications of our results in the context of prior works, and highlight their limitations as well as remaining open questions. In particular, we comment on possibilities for future experimental studies of sampling in biological spiking networks, and applications to neuromorphic computing.

## 2   Spiking networks for sampling-based probabilistic inference

We begin by proposing a framework for probabilistic inference in spiking neural networks in which the spiking rule implements a Metropolis-Hastings step. We show that EBNs [20, 45–49] can be recovered as a limiting case of this more general framework.

In this work, we will keep our discussion quite general, and state our results for sampling from a generic Gaussian distribution. However, the problem we aim to solve can be given a concrete interpretation in a neuroscience context, and could also be extended to non-Gaussian distributions. The goal of a neural network performing probabilistic inference is to estimate a posterior distribution $P(\boldsymbol{\theta}|\mathbf{x})$ over $n_p$ latent variables (or parameters) $\boldsymbol{\theta}$ given an input $\mathbf{x}$ (Figure 1). The input could correspond to the activity of sensory neurons in early sensory processing (e.g. input onto ganglion cells in the retina or onto mitral cells in the olfactory bulb) or inputs into a cortical column that linearly sense features in the environment through an affinity matrix. We provide a detailed discussion of this linear Gaussian model in Appendix C. In the rest of the paper, we will usually abbreviate the distribution from which we want to sample as $P(\boldsymbol{\theta})$, rather than writing $P(\boldsymbol{\theta}|\mathbf{x})$.

## 2.1 Deriving an approximate Metropolis-Hasting spiking sampler

We will build a network of $n_n$ spiking neurons to approximately sample an $n_p$-dimensional Gaussian distribution of time-varying mean $\boldsymbol{\theta}(t)$ and fixed covariance $\boldsymbol{\Psi}$. We first consider the case in which $\boldsymbol{\theta}$ is constant, and then generalize the resulting algorithm to the case in which it is slowly time-varying.

As in prior work on probabilistic inference using spiking networks, we take the samples $\mathbf{z}$ to be linearly decoded from the filtered spike trains $\mathbf{r}$ of $n_n$ neurons [20, 45–49]. Working in discrete time for convenience and clarity (as in [48]), we let

$$\mathbf{z}_t = \boldsymbol{\Gamma}\mathbf{r}_t, \quad \text{where} \quad \mathbf{r}_t = (1 - \eta)\mathbf{r}_{t-1} + \mathbf{o}_t \tag{1}$$

is the low-pass filtered history of spikes $\mathbf{o}_t \in \{0, 1\}^{n_n}$ for some decay constant $0 \leq \eta \leq 1$.

Metropolis-Hastings sampling constructs a Markov chain by drawing a proposed next state from some distribution, and then deciding whether to accept or reject that proposal based on a probabilistic rule [27, 50]. The acceptance ratio is given in terms of the relative posterior probability of the proposed and current states. Here, we will randomly propose which neuron spikes at a given timestep.

In trying to build a Metropolis-Hastings sampler [27, 50] using a probabilistic spiking rule, we are immediately faced with two problems. First, the spikes are sign-constrained and discrete. Second, the dynamics of the filtered spike history incorporate a decay term, hence the readout $\mathbf{z}$ will change even if no spikes are emitted. These conditions mean that the proposal density over $\mathbf{z}$ will not be symmetric, and that the Markov process will not satisfy the condition of detailed balance [27, 50]. We note that previous work has shown that violations of detailed balance can accelerate sampling [17, 18, 51], but we will not carefully explore this possibility in the present work. To obtain a symmetric proposal distribution with sign-constrained spikes, we assume that the network is divided into two equally-sized populations with equal and opposite readout weights, i.e. that the readout matrix is of the form $\boldsymbol{\Gamma} = [+\mathbf{M}, -\mathbf{M}]$ for some matrix $\mathbf{M} \in \mathbb{R}^{n_p \times n_n/2}$. This could be accomplished by dividing the total population of neurons into excitatory and inhibitory populations with weights that are fine-tuned to be equal and opposite. The second problem can be solved by assuming that $\eta = 0$, i.e., that we have access to a perfect integrator of the spike trains.

At the $t$-th timestep, we choose one neuron $j$ uniformly at random, and let the spike proposal be $\mathbf{o}' = \mathbf{e}_j$, where $\mathbf{e}_j$ is the $j$-th standard Euclidean basis vector (i.e., $(\mathbf{e}_j)_k = \delta_{jk}$). This yields a candidate readout

$$\mathbf{z}' = (1 - \eta)\mathbf{z}_{t-1} + \boldsymbol{\Gamma}\mathbf{e}_j. \tag{2}$$

If $\eta = 0$ and the balance assumption on $\boldsymbol{\Gamma}$ is satisfied, then the proposal distribution is exactly symmetric, in the sense that the probabilities of reaching $\mathbf{z}'$ from $\mathbf{z}_{t-1}$ and of reaching $\mathbf{z}_{t-1}$ from $\mathbf{z}'$ are equal. Then, if we accept the proposed spike with probability

$$A = \min\left\{1, \frac{P(\mathbf{z}')}{P(\mathbf{z}_{t-1})}\right\}, \tag{3}$$

we obtain a Metropolis-Hastings sampling algorithm for the discretized Gaussian [27, 50].

We now relax the assumption of the perfect integration, and assume only that $\eta \ll 1$. Introducing decay smooths the readout, which in the perfect integrator is discretized. With the same proposal distribution as before, we take the acceptance ratio of the accept-reject step to be

$$A = \min\left\{1, \frac{P[(1 - \eta)\mathbf{z}_{t-1} + \boldsymbol{\Gamma}\mathbf{e}_j]}{P[(1 - \eta)\mathbf{z}_{t-1}]}\right\}. \tag{4}$$

This choice has two important features. First, the decay means that the proposal distribution will be asymmetric, and the Markov chain will no longer satisfy the condition of detailed balance [27, 50]. However, the resulting error will be small if $\eta \ll 1$. Second, by comparing the likelihood of the proposal, $P[(1 - \eta)\mathbf{z}_{t-1} + \mathbf{\Gamma}\mathbf{e}_j]$, to the likelihood of the next state without the proposed spike but with the decay $P[(1 - \eta)\mathbf{z}_{t-1}]$ (instead of the likelihood of the current state $P[\mathbf{z}_{t-1}]$ as in the Metropolis-Hastings algorithm), this choice implements a sort of look-ahead step that should allow the algorithm to partially compensate for the decay in the rate.

With the choices above, we show in Appendix B that one can write the acceptance ratio (4) as

$$A = \min \left\{ 1, \exp\left(V_{t-1,j} - T_j\right) \right\}, \tag{5}$$

where $\mathbf{V}_{t-1} = -(1 - \eta)\mathbf{\Omega}\mathbf{r}_{t-1} + \mathbf{\Gamma}^\top \mathbf{\Psi}^{-1}\boldsymbol{\theta}$ has the interpretation of a membrane potential,

$$T_j = \frac{1}{2}\Omega_{jj} \tag{6}$$

has the interpretation of a spiking threshold, and the the recurrent weight matrix is defined as

$$\mathbf{\Omega} \equiv \mathbf{\Gamma}^\top \mathbf{\Psi}^{-1}\mathbf{\Gamma}. \tag{7}$$

Thus far, we have assumed that the mean signal is constant. The natural generalization of this algorithm to a time-varying mean signal $\boldsymbol{\theta}_t$ is to take the membrane potential to be $\mathbf{V}_t = -(1 - \eta)\mathbf{\Omega}\mathbf{r}_t + \mathbf{\Gamma}^\top \mathbf{\Psi}^{-1}\boldsymbol{\theta}_t$. This leads to the voltage dynamics

$$\mathbf{V}_t - (1 - \eta)\mathbf{V}_{t-1} = -(1 - \eta)\mathbf{\Omega}\mathbf{o}_t + \mathbf{\Gamma}^\top \mathbf{\Psi}^{-1}[\boldsymbol{\theta}_t - (1 - \eta)\boldsymbol{\theta}_{t-1}], \tag{8}$$

which, when combined with the probabilistic spiking rule with uniform proposals and acceptance ratio (5), yields our final algorithm. This will not be an exact Metropolis-Hastings sampler unless the mean is constant, the decay term vanishes ($\eta = 0$), and the readout matrix $\mathbf{\Gamma}$ satisfies an exact balancing condition. In particular, if these conditions are violated, the Markov chain will not satisfy detailed balance. However, if they are violated only weakly, this algorithm can still be a reasonable approximation to a true sampler. We will provide empirical evidence for this intuition in §3.

## 2.2 The continuous-time limit

We now consider the continuous-time limit of the model introduced above. This limit corresponds to taking the limit in which spike proposals are made infinitely often, and regarding the dynamics written down previously as a forward Euler discretization of an underlying continuous-time system. For a timestep $\Delta$ between spike proposals, we let the discrete-time decay rate be $\eta = \Delta/\tau_m$ for a time constant $\tau_m$, thusly named because it has the interpretation of a membrane time constant. Then, we show in Appendix B that the $\Delta \downarrow 0$ limit of the discretized rate dynamics (1) yields the familiar continuous-time dynamics

$$\frac{d\mathbf{r}(t)}{dt} = -\frac{1}{\tau_m}\mathbf{r}(t) + \mathbf{o}(t). \tag{9}$$

In continuous time, the spike train $\mathbf{o}(t)$ is now composed of Dirac delta functions, as the discretized spikes are rectangular pulses of width $\Delta$ in time and height $1/\Delta$. We next consider the voltage dynamics of the leaky integrator for a varying mean signal (8), which have a similar continuum limit:

$$\frac{d\mathbf{V}(t)}{dt} = -\frac{1}{\tau_m}\mathbf{V}(t) - \mathbf{\Omega}\mathbf{o}(t) + \mathbf{\Gamma}^\top \mathbf{\Psi}^{-1}\left(\frac{d\boldsymbol{\theta}(t)}{dt} + \frac{1}{\tau_m}\boldsymbol{\theta}(t)\right). \tag{10}$$

In this limit, the rate will decay by only a infinitesimal amount between a rejected spike proposal and the next proposal, meaning that the error incurred by neglecting the asymmetry in the proposal distribution due to the decay should be negligible. We also note that, though the probabilistic spike rule (5) does not explicitly include a reset step, the dynamics (10) prescribe that the $j$-th neuron's membrane voltage should be decremented by $2T_j$ after it spikes.

## 2.3 Efficient balanced networks as a limit of the spiking Metropolis-Hastings sampler

This spiking network samples the posterior distribution by emitting spikes probabilistically, but we can use the same architecture to re-derive EBNs, which approximate continuous dynamical systems

using spiking networks [45–49]. If we take $\boldsymbol{\Psi} \propto \mathbf{I}_{n_p}$ (for $\mathbf{I}_{n_p}$ the $n_p \times n_p$ identity matrix) the voltage dynamics (10) are identical to those of the EBN.[3] The greedy spiking rule of the EBN can be recovered in this framework by taking a limit in which the variance of the Gaussian target distribution vanishes. Concretely, we let $\boldsymbol{\Psi} = \psi \mathbf{I}_{n_p}$, and define re-scaled variables $\tilde{\mathbf{V}}(t) \equiv \psi \mathbf{V}(t)$, $\tilde{\boldsymbol{\Omega}} \equiv \psi \boldsymbol{\Omega}$, and $\tilde{\mathbf{T}} \equiv \psi \mathbf{T}$ that will remain $\mathcal{O}(1)$ even as we take $\psi \downarrow 0$. In terms of these new variables, the acceptance ratio is $A = \min\{1, \exp[(\tilde{V}_j(t) - \tilde{T}_j)/\psi]\}$, which tends to $A = \Theta(\tilde{V}_j(t) - \tilde{T}_j)$ as $\psi \downarrow 0$. This explicitly recovers the greedy spike rule used in EBNs. In Appendix B, we further analyze how the overall scales of $\boldsymbol{\Psi}$ and $\boldsymbol{\Gamma}$ affect the probabilistic spike rule.

The network with voltage dynamics (10) samples a distribution with mean $\boldsymbol{\theta}$ and covariance $\boldsymbol{\Psi}$. Instead of sampling using the structured proposal distribution on spikes, this network can implement sampling through slowly varying Langevin dynamics on $\boldsymbol{\theta}$. In the limit where the spike rule becomes greedy, this recovers the spiking sampler studied by Savin and Denève [20]. We will discuss this model further in §3.

## 3 Population geometry for fast sampling

### 3.1 Leveraging the geometry of inference to accelerate sampling

We first review recent work from the machine learning literature for how population geometry can be chosen to accelerate simple Langevin sampling, which establishes principles for the design of fast samplers. For probability distributions belonging to the exponential family, including the Gaussian distributions on which we focus in this work, one can write the density $P(\mathbf{z})$ in terms of an energy function $U(\mathbf{z})$ such that $P(\mathbf{z}) \propto \exp[-U(\mathbf{z})]$. The classic algorithm to sample such a distribution is the discretization of the naïve Langevin dynamics

$$d\mathbf{z}(t) = -\boldsymbol{\nabla} U(\mathbf{z}) \, dt + \sqrt{2} \, d\mathbf{W}(t), \qquad (11)$$

where $\mathbf{W}(t)$ is a standard Brownian motion [29, 30, 33, 52]. By simply following these dynamics, one can obtain samples from the target distribution and therefore an estimate of the uncertainty at the timescale taken by the network to sufficiently explore the target distribution.

The Langevin dynamics (11) can be directly implemented in a rate network [9, 19] or approximately implemented in a spiking network [20], but their convergence properties for high-dimensional distributions have not been investigated in a neuroscience context. It is well known in statistics that, as the dimensionality of the target distribution increases, convergence of Langevin sampling to the target distribution slows dramatically. Furthermore, the discretization step can induce errors that cause the variance estimated from sampling to exceed the target variance [29, 30, 33, 52].

To overcome these issues, prior work in statistics and machine learning has proposed algorithms that can accelerate the sampling [28, 31, 42–44, 53–55] which we leverage here to propose our fast samplers in spiking neural networks. These ideas were unified into a common framework by Ma et al. [43] (and see also [44, 56]) who proposed a "complete recipe" for stochastic gradient MCMC:

$$d\mathbf{z}(t) = -\{[\mathbf{D}(\mathbf{z}) + \mathbf{S}(\mathbf{z})]\boldsymbol{\nabla} U(\mathbf{z}) + \boldsymbol{\Phi}(\mathbf{z})\} \, dt + \mathbf{B}\sqrt{2} \, d\mathbf{W}(t) \qquad (12)$$

with $\mathbf{B}\mathbf{B}^\top = \mathbf{D}$. The matrix fields $\mathbf{D}(\mathbf{z})$ and $\mathbf{S}(\mathbf{z})$ modify the dynamics but keep the target distribution unchanged. $\mathbf{D}$ is positive semi-definite and defines the local geometry of the space in which the inference is occurring, while $\mathbf{S}$ is skew-symmetric and adds non-reversible dynamics. When $\mathbf{D}$ and $\mathbf{S}$ are state dependent, the correction term $\boldsymbol{\Phi} = \mathrm{div}(\mathbf{D} + \mathbf{S})$ must be included [43].

The "complete recipe" provides a general framework to design samplers based on Langevin dynamics. Samplers based on Riemannian geometry can be designed by choosing $\mathbf{D}$ to be the inverse of the Fisher information matrix $\mathbf{G}$ (or an approximation thereof), yielding the natural gradient $\boldsymbol{\nabla}_{\mathrm{nat}} U = \mathbf{G}^{-1}\boldsymbol{\nabla} U$ (Figure 1) [39–42, 57, 58]. Samplers incorporating dummy variables can be designed by expanding the parameter space and using the matrices $\mathbf{D}$ and $\mathbf{S}$ to obtain the desired dynamics [31, 43, 44, 53–55, 59]. Similarly, prior works have proposed methods to accelerate sampling

---

[3]In Appendix B, we show how the elastic net prior on firing rates used by Boerlin et al. [45] can be incorporated into this model. As it only modifies the definitions of the recurrent weights and spiking threshold, this extension does not add new conceptual difficulties, hence we do not discuss it further in the main text. Additionally, we provide a pedagogical introduction to the dynamics of the EBN in Appendix C.

in biologically inspired neural networks by parallelizing the inference [20], using Hamiltonian dynamics [21] or by adding non-reversible dynamics [9, 17, 18]. The "complete recipe" provides a general framework that encompasses all these examples, allowing for the principled design of biologically-plausible samplers.

In the simple setting of linear rate networks sampling from a Gaussian distribution with constant mean $\boldsymbol{\mu}$ and covariance $\boldsymbol{\Sigma}$, the complete recipe for state-independent $\mathbf{D}$ and $\mathbf{S}$ takes the form

$$d\mathbf{z}(t) = -(\mathbf{D} + \mathbf{S})\boldsymbol{\Sigma}^{-1}(\mathbf{z} - \boldsymbol{\mu})\, dt + \sqrt{2}\mathbf{B}\, d\mathbf{W}(t). \tag{13}$$

In Appendix D, we show in the zero-mean case $\boldsymbol{\mu} = \mathbf{0}$ that large eigenvalues of the covariance matrix $\boldsymbol{\Sigma}$ introduce long timescales in the sample cross-covariance for naïve Langevin dynamics ($\mathbf{D} = \mathbf{B} = \mathbf{I}_{n_p}$, $\mathbf{S} = \mathbf{0}$), which slow convergence to the target distribution. Concretely, the 2-Wasserstein distance between the ensemble sampling distribution at time $t$ and the target is

$$W_2^{\text{naïve}}(t) = \sqrt{\sum_{i=1}^{n_p} \sigma_i \left[1 - (1 - e^{-2t/\sigma_i})^{1/2}\right]^2}, \tag{14}$$

where $\sigma_i$ are the eigenvalues of the covariance matrix. Networks performing inference in the natural space ($\mathbf{D} = \boldsymbol{\Sigma}$, $\mathbf{B} = \boldsymbol{\Sigma}^{1/2}$) are insensitive to these large eigenvalues, with 2-Wasserstein distance

$$W_2^{\text{natural}}(t) = \sqrt{\sum_{i=1}^{n_p} \sigma_i \left[1 - (1 - e^{-2t})^{1/2}\right]}. \tag{15}$$

For this analytical result, we consider the distribution of samples at time $t$ over realizations of the noise process, which differs from the distribution of samples over time within a single realization, as used elsewhere in the paper. This choice is made because the joint distribution of samples within a single realization is challenging to characterize [60]. These results show how natural gradient can accelerate inference in linear rate networks, and complement Hennequin et al. [9]'s study of how adding irreversible dynamics can reduce the sample autocorrelation time (i.e., $\mathbf{D} = \mathbf{B} = \mathbf{I}_{n_p}$, $\mathbf{S} \neq \mathbf{0}$).

## 3.2   Fast sampling through population geometry in efficient balanced networks

We first consider a sampler based on efficient neural networks [20, 45–47] that leverages the geometry of the inference to implement efficient sampling at the level of the population. In previous work, Savin and Denève [20] derived a sampler implementing naïve Langevin dynamics (we provide a full derivation using the notation from the present paper in Appendix C). Although they proposed to accelerate sampling by implementing parallel inference loops, they do not leverage the geometry of the inference nor do they test the convergence in high ($n_p > 2$) dimensions.

Here, we approximate the "complete recipe" dynamics (13) using an EBN, and show that performing inference using natural gradients helps with speed and accuracy in high dimensions. As in §3, we consider sampling from a Gaussian distribution of possibly time-varying mean $\boldsymbol{\mu}$ and covariance $\boldsymbol{\Sigma}$.[4] As we principally study the effect of the geometry, we henceforth set $\mathbf{S} = \mathbf{0}$. Note that even though the underlying dynamics of (13) are reversible if $\mathbf{S} = \mathbf{0}$, the non-linearities introduced by spiking lead to a non-reversible sampler. To embed these dynamics in the EBN framework, we take the target mean in the voltage dynamics (10) to evolve according to the Ornstein-Uhlenbeck process (13), which we generalize to include an intrinsic timescale $\tau_s$. Following [20], we approximate the true state of the sampling dynamics by the instantaneous estimate from the filtered spike history. Then, as detailed in Appendix C, we obtain an EBN with the voltage dynamics

$$d\mathbf{V} = \left[-\frac{1}{\tau_m}\mathbf{V} - \boldsymbol{\Omega}\mathbf{o} + \frac{1}{\tau_m}\boldsymbol{\Gamma}^\top \left(\mathbf{I}_{n_p} - \frac{\tau_m}{\tau_s}\mathbf{D}\boldsymbol{\Sigma}^{-1}\right)\boldsymbol{\Gamma}\mathbf{r} + \frac{1}{\tau_s}\boldsymbol{\Gamma}^\top \mathbf{D}\boldsymbol{\Sigma}^{-1}\boldsymbol{\mu}\right] dt + \sqrt{\frac{2}{\tau_s}}\boldsymbol{\Gamma}^\top \mathbf{B}\, d\mathbf{W}, \tag{16}$$

where the recurrent weight matrix is $\boldsymbol{\Omega} = \boldsymbol{\Gamma}^\top \boldsymbol{\Gamma}$, and the spiking rule is greedy, with thresholds $T_j = (\boldsymbol{\Gamma}^\top \boldsymbol{\Gamma})_{jj}/2$.

---

[4]In Appendix C, we give a concrete interpretation of this task in terms of the Gaussian linear models used in previous work by Hennequin et al. [9] and Savin and Denève [20].

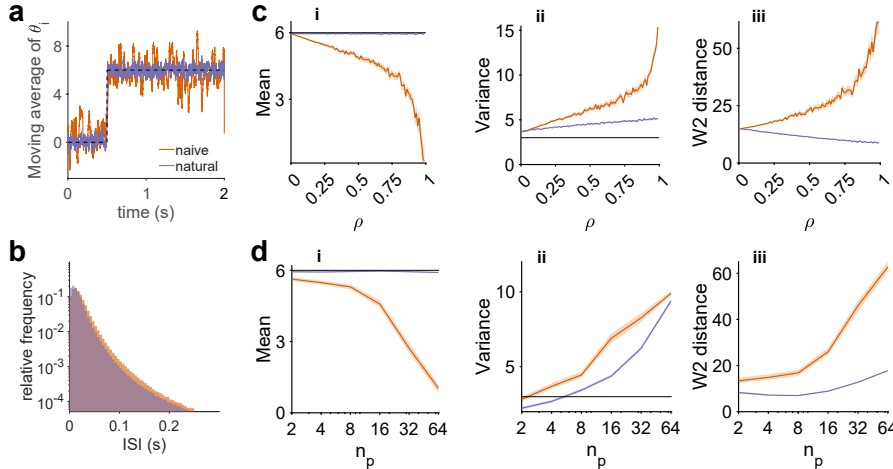

Figure 2: Accelerating inference through population geometry in EBNs simulating Langevin sampling:
**a**. 50 ms moving averages of parameter estimates $\hat{\theta}_i$ over time, for naïve and natural geometry in a network sampling from $n_p = 20$-dimensional equicorrelated Gaussian distribution of correlation $\rho = 0.75$ using a network of $n_n = 200$ neurons. At $t = 0.5$ s, the target mean shifts from zero to six, as indicated by the black dashed line. **b**. The distribution of ISIs across neurons and trials is approximately exponential. See Supplemental Figure E.2 for more statistics of the resulting spike trains. **c**. Comparison of performance in the 50 ms following stimulus onset (after $t = 0.5$ s) between naïve and natural geometry for varying $\rho$ **c.i**. The estimate of the mean collapses towards zero with increasing $\rho$ for naïve geometry, **c.ii**. The estimated variance increases catastrophically with $\rho$ for naïve geometry, but only mildly for natural geometry. **c.iii**. Inference accuracy as measured by the mean marginal 2-Wasserstein distance across dimensions decreases for naïve. **d**. Similar analysis when varying $n_p$ for the **d.i**. Mean, **d.ii**. Variance and **d.iii**. mean 2-Wasserstein distance. In panels **c,d** shaded error patches show 95% confidence intervals computed via bootstrapping over 100 realizations in all panels. In Supplemental Figure E.1 we show the full time course of the inferred statistics for one set of network parameters and in Supplemental Figure E.3 we show the statistics after the full 1.5s, which are qualitatively similar. See Appendix E for detailed numerical methods.

To illustrate how correlations between parameters affect sampling in high dimensions, we will focus on equicorrelated multivariate distributions. The covariance matrix $\mathbf{\Sigma}$ of such a distribution is parameterized by an overall variance $\sigma > 0$ and a correlation coefficient $-1 < \rho < 1$ such that $\Sigma_{ij} = \sigma[\delta_{ij} + \rho(1 - \delta_{ij})]$. We will explore the performance of the sampling algorithms across values of $\rho$ for different dimensions of the parameter ($n_p$) and neuron ($n_n$) spaces. For a multivariate Gaussian distribution $\mathcal{N}(\boldsymbol{\mu}, \mathbf{\Sigma})$, the Fisher information matrix is $\mathbf{G} = \mathbf{\Sigma}^{-1}$ and we will therefore use $\mathbf{D} = \mathbf{G}^{-1} = \mathbf{\Sigma}$ (and $\mathbf{B} = \sqrt{\mathbf{D}}$) in our geometry aware implementation (see Appendix C). In our simulations, we compare the accuracy of sampling of the naïve implementation ($\mathbf{D} = \mathbf{B} = \mathbf{I}$) with the geometry-aware version over a 50 millisecond window, which is roughly twice the membrane time constant $\tau_m = 20$ ms, as well as at steady-state. In Figure 2 and in Supplemental Figures E.1-E.3, we show that the geometry-aware sampler is more robust to increasing the correlation of the parameters and the dimensionality, allowing inference at behavioral timescales. As in non-spiking Langevin sampling, discretization leads to an overestimation of variance [27], but this effect, although still present, is strongly reduced in the geometry-aware implementation. Note here, that the geometry is imposed through the dynamics of the membrane potentials via the recurrent connectivity in the network.

### 3.3 Fast sampling through population geometry using probabilistic spike rules

In the preceding subsection, we have shown how neural population geometry, implemented through neural dynamics, can accelerate the speed of approximate sampling in an efficient balanced network. However, this approach suffers from a fundamental conceptual gap: the firing rates of the spiking network are being used to simulate non-spiking Langevin dynamics; the spiking network itself has not been designed to sample. Specifically, the discretization introduced by the spiking exacerbates

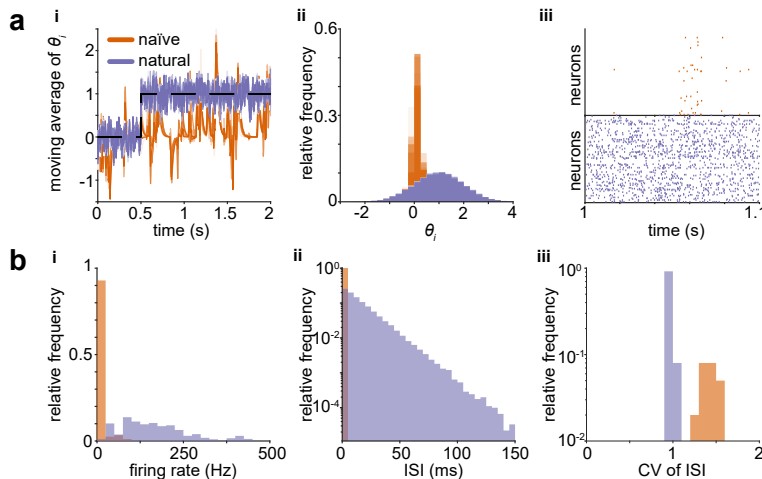

Figure 3: Sampling using probabilistic spiking rules. **a**. Sampling from a $n_p = 10$-dimensional equicorrelated Gaussian distribution of correlation $\rho = 0.75$ using a network of $n_n = 100$ neurons. **a.i**. 100 ms moving averages of parameter estimates $\hat{\theta}_i$ over time, for naïve and natural geometry. At $t = 0.5$ s, the target mean shifts from zero to one, as indicated by the black dashed line. **a.ii**. Marginal distributions of $\theta_i$ after stimulus appearance. **a.iii** Spike rasters over a 100 ms window. **b**. Variability in spiking across 1000 trials. **b.i**. Spike rate distributions across neurons and trials for a stimulus distribution as in (a). **b.ii**. The distribution of ISIs for an example neuron across trials is approximately exponential. **b.iii**. The distribution of coefficients of variation of ISIs across neurons. See Appendix E for detailed numerical methods.

the errors introduced by the discrete time implementation of the sampling dynamics, leading to overestimation of the stimulus variance. In this subsection, we take a first step towards bridging this gap by considering an alternative limit of the general model introduced in §2: the case in which sampling is performed leveraging only the stochasticity in the spike rule. Our objective here is not to demonstrate a fully biologically-plausible or practically useful sampling algorithm; rather, it is to illustrate the importance of population geometry in a minimal model for probabilistic spiking.

As in 3.2, we focus on sampling from equicorrelated Gaussian distributions $\mathcal{N}(\boldsymbol{\mu}, \boldsymbol{\Sigma})$. In this case, we set $\boldsymbol{\Psi} = \boldsymbol{\Sigma}$ and $\boldsymbol{\theta} = \boldsymbol{\mu}$ in (10). Naïve sampling here corresponds to choosing $\boldsymbol{\Gamma}$ in some sense generically (see Appendix B), while geometry-aware sampling corresponds to choosing $\boldsymbol{\Gamma}$ such that $\boldsymbol{\Gamma}\boldsymbol{\Gamma}^\top \simeq \boldsymbol{\Sigma}$ up to overall constants of proportionality. In Figures 3 and 4, and in Supplemental Figure E.4, we show that naïve choices of $\boldsymbol{\Gamma}$ lead to vanishing spike rates at strong correlations $\rho$ and large parameter-space dimensionalities $n_p$. This results in dramatic underestimation of the mean and variance of the target distribution, which is resolved by choosing the geometry appropriately, again allowing inference at behavioral time-scales. Moreover, these networks show Poisson-like variability in spiking statistics (Figure 3), consistent with cortical dynamics [61, 62].

In Appendix B, we provide a more careful analysis of the strongly-correlated limit $\rho \uparrow 1$. Informally, we show that the probability of spiking should vanish if $\boldsymbol{\Gamma}$ is chosen sufficiently naïvely and the mean of the target distribution is uniform across parameter dimensions, i.e., $\boldsymbol{\mu}_t \propto \mathbf{1}_{n_p}$. This analysis is not specialized to a particular case, and holds generally for the model of §2. Therefore, careful choice of population geometry, as implemented by the neural code, is required for fast sampling in this model.

## 4    Discussion

In this paper, we have shown how careful choice of neural population geometry enables fast sampling in spiking neural networks. We presented a unified framework in which EBN samplers approximating Langevin dynamics with greedy spiking and approximate Metropolis-Hastings samplers with deterministic voltage dynamics and probabilistic spiking can be unified. We then leveraged population geometry to perform rapid sampling at behaviorally-relevant timescales in these two disparate limits of our general model. We now discuss some of the limitations of our work, and highlight possible directions for future inquiry.

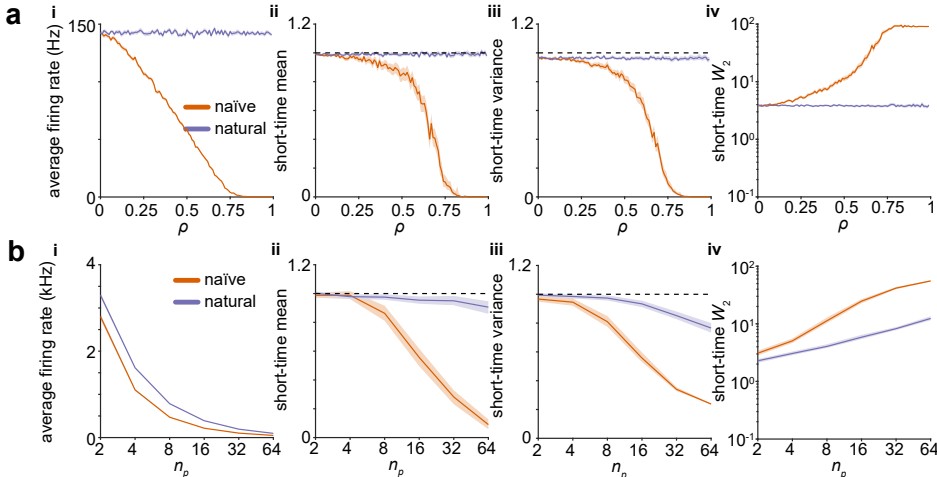

Figure 4: Population geometry enables fast sampling from strongly-correlated high-dimensional distributions in networks with probabilistic spiking rules. **a**. Sampling from strongly-equicorrelated Gaussians in $n_p = 10$ dimensions using a network of $n_n = 100$ neurons requires careful choice of geometry. The stimulus setup is as in Figure 3. **a.i**. At strong correlations $\rho$, spiking is suppressed in networks with naïve geometry, but not when the natural geometry is used. **a.ii**. Dimension-averaged estimate of the mean signal in the 50 ms after stimulus onset. See Supplemental Figure E.4 for steady-state statistics. **a.iii**. As in **ii**, but for the variance. **a.iv**. As in **ii**, but for the mean marginal 2-Wasserstein distance across dimensions. **b**. Sampling in high dimensions requires careful choice of geometry. **b.i**. At moderately strong correlations $\rho = 0.75$ and high dimensions, spiking is suppressed in networks with naïve geometry, but not when the natural geometry is used. Here, we use 10 neurons per parameter, i.e., $n_n = 10 n_p$. **b.ii**. Dimension-averaged estimate of the mean signal in the 50 milliseconds after stimulus onset. See Supplemental Figure E.4 for steady-state statistics. **b.iii**. As in **ii**, but for the variance. **b.iv**. As in **ii**, but for the mean 2-Wasserstein distance. Shaded error patches show 95% confidence intervals computed via bootstrapping over 100 realizations in all panels; see Appendix E for further details.

Like the original EBN model, the probabilistic spiking model introduced in §2 suffers from the limitations that it requires instantaneous propagation of spike information and that only one neuron is allowed to spike at a time [20, 45–49]. Moreover, the discretization timestep enforces a hard cutoff on the maximum spike rate. Some of these limitations could be partially circumvented by generalizing the spike proposal distribution to allow multiple spikes. However, such a model would still suffer from the issue that an accept/reject step that accounts for the effect of spikes from multiple neurons will require instantaneous communication across the network. This limitation could possibly be overcome within the framework of asynchronous Gibbs sampling [63, 64], which ignores the requirement that updates should be coordinated across the network.

The analysis of §2 shows that the models of §3 can be viewed as limiting cases of a single framework. It is likely that the parallels between these limiting models could be further strengthened by viewing the Gaussian noise in the voltage dynamics of the EBN sampler as an approximation to the effect of the stochastic spiking of other neurons in the Metropolis-Hastings sampler. Such an approximation could be obtained in the limit of large network size given an appropriate treatment of asynchronous updates, provided that one could neglect the coupling between spikes in different neurons induced by the proposal distribution [62, 65–67]. Careful analysis of the relationship between these two sources of stochasticity will be an interesting subject for future investigation. Moreover, it will be interesting to investigate how they might be integrated in a single network, which would result in a spiking sampler somewhat reminiscent of the Metropolis-adjusted Langevin algorithms used in machine learning [68, 69].

Here, we have focused entirely on Gaussian target distributions. As we sketch in Appendices B and C, both the probabilistic spiking sampler and the EBN sampler could in principle be extended to general exponential family targets. However, the most naïve extensions to non-Gaussian distributions would involve non-linear and non-local interactions in the voltage dynamics. In particular, one would have to account for state-dependence in the matrix fields $\mathbf{D}(\mathbf{z})$ and $\mathbf{S}(\mathbf{z})$, which would require the inclusion of

the field $\mathbf{\Phi}(\mathbf{z})$ in the complete recipe (12) (see ref. [43]). Mapping such interactions onto biological mechanisms requires more careful consideration. In recent work, Nardin et al. [70] have proposed how dynamical systems with polynomial nonlinearities can be approximated by deterministic EBNs with multiplicative synapses. This approach could be extended to the stochastic setting of networks designed to sample, which will be interesting to test in future work.

In this work, we did not constrain neurons to follow Dale's law, and single neurons therefore have both excitatory and inhibitory effects on their neighbors. Many frameworks have been proposed to map the connectivity of unconstrained network algorithms onto distinct excitatory and inhibitory neuron types [9, 71, 72]. These refinements of the biological plausibility will not affect our key argument of accelerating inference through a favorable population geometry. However, different possible implementations that comply with Dale's law will make different predictions for experimentally-measureable biophysical properties. For example, although less numerous, fast spiking inhibitory neurons have higher firing rates [71, 72], which could allow the approximate symmetry of the readout, as required by the construction of §2, to be maintained.

In biological spiking networks, probabilistic spike emission and probabilistic synaptic release are natural sources of stochasticity [73–76]. These two layers of probabilistic computation provide additional flexibility in processing beyond the simple accept/reject step considered here. As a result, it is likely that one could construct sampling algorithms that are at once more biophysically detailed and more computationally efficient than the simple network constructed in §2. Further investigation of how violations of detailed balance through these mechanisms and the matrix $\mathbf{S}$ in the "complete recipe" framework could enable faster sampling will be a particularly interesting objective [17, 18, 51]. Moreover, it will be interesting to investigate how natural gradients can accelerate inference in the presence of more complex synaptic dynamics, as studied in recent work by Kreutzer et al. [77].

The algorithms proposed in this work could also enable fast sampling in neuromorphic circuits [78–80]. As they require only the local membrane voltage to compute accept/reject steps, these algorithms could be implemented in a distributed neuromorphic architecture. They would then potentially be limited only by the timescale of individual computing units (which are much faster than biological neurons) rather than the dimensionality of the inference problem [81].

The sampling processes considered in this work focus on short-timescale perceptual inference. However, similar probabilistic inference can occur at longer timescales of learning [82–84]. These long-timescale sampling processes would allow networks to flexibly infer their synaptic weights. Importantly, learning the task—i.e., learning the matrix $\mathbf{\Sigma}^{-1}$—and learning the representational geometry—$\mathbf{D}$, $\mathbf{S}$, and $\mathbf{\Gamma}$—are distinct processes that can occur in parallel. Once the task structure is learned, adapting the geometry of the population code to match changing stimulus geometry can be achieved through meta-learning [85]. Recent works have analytically studied the population geometry that results from this sampling procedure in rate-based networks [86], and proposed algorithms for efficient learning in EBNs [47, 87–89] and other classes of spiking networks [90–93]. However, the interactions between fast activity sampling and slow network parameter sampling in neural circuits—particularly spiking networks—remain poorly understood [84]. Characterizing how adaptive population geometry accelerates perceptual inference in dynamic environments will be an important step towards a more complete understanding of probabilistic inference in neural circuits.

## Acknowledgments and Disclosure of Funding

PM was supported by the Harvard Mind Brain Behavior Interfaculty Initiative. JAZ-V and CP were supported by a Google Faculty Research Award and NSF Award DMS-2134157. This work was also supported by a grant from the National Institute of Health (R01-DC-017311) to VNM. We thank Tony Zador for co-organizing the NeurIPS 1996 workshop on "Synaptic transmission: reliability and variability", whose follow-up discussions led to this paper.

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
