# Supplemental material for "Natural gradient enables fast sampling in spiking neural networks"

**Paul Masset**[1,2]*, **Jacob A. Zavatone-Veth**[1,3]*, **J. Patrick Connor**[4],
**Venkatesh N. Murthy**[1,2]†, **Cengiz Pehlevan**[1,4]†

[1]Center for Brain Science, [2]Department of Molecular and Cellular Biology,
[3]Department of Physics, [4]John A. Paulson School of Engineering and Applied Sciences,
Harvard University
Cambridge, MA 02138
`paul_masset@fas.harvard.edu, jzavatoneveth@g.harvard.edu,`
`cpehlevan@seas.harvard.edu`

## Contents

---

*PM and JAZ-V contributed equally to this work.
†VNM and CP jointly supervised this work.

36th Conference on Neural Information Processing Systems (NeurIPS 2022).

## A   Table of variables and parameters used in the models and simulations

We provide a table stating the dimensionality and values taken by the variables and parameters used throughout the paper.

Table 1: Variable and parameter names

| Variable name | Description | Value/Size |
|---|---|---|
| $n_p$ | Number of features encoded by the network | $2 - 128$ |
| $n_n$ | Number of neurons in the network | $2 - 2560$ |
| $n_i$ | Dimensionality of the input signal | usually $n_p$ |
| $\boldsymbol{\theta}$ | Features inferred by the network | $n_p \times 1$ |
| $\boldsymbol{\Gamma}$ | Readout weights | $n_p \times n_n$ |
| $\boldsymbol{\Psi}$ | Covariance of structured Gaussian posterior | $n_p \times n_p$ |
| $\mathbf{V}$ | Membrane voltage | $n_n \times 1$ |
| $\mathbf{T}$ | Spiking thresholds (for each neuron) | $n_n \times 1$ |
| $\boldsymbol{\Omega}$ | Recurrent weights | $n_n \times n_n$ |
| $\tau_m$ | Membrane time constant | scalar; $\sim 20$ ms |
| $\Delta$ | Discretization timestep | scalar |
| $\eta$ | Discrete-time decay constant; $\eta = \Delta/\tau_m$ | scalar |
| $\alpha$ | Elastic net $\ell_1$ penalty | scalar |
| $\lambda$ | Elastic net $\ell_2$ penalty | scalar |
| $\boldsymbol{\mu}$ | Mean of target Gaussian distribution | $n_p \times 1$ |
| $\boldsymbol{\Sigma}$ | Covariance of target Gaussian distribution | $n_p \times n_p$ |
| $\rho$ | Cross-correlation of equicorrelated Gaussian | [0-0.99] |
| $\mathbf{x}$ | Input signal | $n_i$ |
| $\mathbf{A}$ | Input weights | $n_p \times n_i$ |
| $\mathbf{W}$ | Standard Brownian motion | usually $n_p \times 1$ |

## B   Approximate Metropolis-Hastings sampling using probabilistic spiking rules

In this Appendix, we provide a step-by-step construction of the spiking sampler introduced in §2 of the main text. Our goal is to use

$$\mathbf{z}_t = \boldsymbol{\Gamma}\mathbf{r}_t \tag{B.1}$$

to sample a Gaussian distribution $P(\mathbf{z})$ with mean $\boldsymbol{\theta}$ and covariance $\boldsymbol{\Psi}$, where

$$\mathbf{r}_t = (1 - \eta)\mathbf{r}_{t-1} + \mathbf{o}_t \tag{B.2}$$

is the filtered spike history for a decay constant $0 \leq \eta \leq 1$ (note that $\mathbf{o}_t \in \{0, 1\}^{n_n}$). In §B.1, we construct a circuit that samples a Gaussian distribution using a discrete-time perfect integrator of the spike train (i.e., $\eta = 0$). In §B.2, we relax this assumption, yielding an approximate sampler using leaky integration in discrete-time, and discuss the behavior of this model in the continuum limit. In general, the proposal distribution could be computed using a stochastic gradient step followed by an accept/reject step, yielding Metropolis-adjusted Langevin dynamics [1, 2]. In this case, the accept/reject step allows the algorithm to compensate for some of the sampling error introduced by discretization at the expense of needing to compute a likelihood ratio, which can be expensive for high-dimensional distributions.

## B.1 A simple sampler assuming perfect integration and balance

We first construct a sampling circuit under the assumptions that we have access to a perfect integrator of the spike train, i.e. that $\eta = 0$, and that the readout matrix is of the form

$$\boldsymbol{\Gamma} = [+\mathbf{M} \quad -\mathbf{M}] \tag{B.3}$$

for some matrix $\mathbf{M} \in \mathbb{R}^{n_p \times n_n/2}$.

At the $t$-th timestep, we choose one neuron $j$ uniformly at random, and let the spike proposal be $\mathbf{o}' = \mathbf{e}_j$, where $\mathbf{e}_j$ is the $j$-th standard Euclidean basis vector (i.e., $(e_j)_k = \delta_{jk}$). This yields a candidate readout

$$\mathbf{z}' = \boldsymbol{\Gamma}(\mathbf{r}_{t-1} + \mathbf{e}_j) = \mathbf{z}_{t-1} + \boldsymbol{\Gamma}\mathbf{e}_j. \tag{B.4}$$

Under the symmetry assumption on $\boldsymbol{\Gamma}$, the acceptance ratio is given by

$$A = \min\left\{1, \frac{P(\mathbf{z}')}{P(\mathbf{z}_{t-1})}\right\}, \tag{B.5}$$

as the proposal distribution is exactly symmetric in $\mathbf{z}'$ and $\mathbf{z}_t$. Then, we accept the proposed spike with probability $A$. Concretely, for $u \sim \mathcal{U}[0, 1]$, we take

$$\mathbf{o}_t = \begin{cases} \mathbf{e}_j & \text{if } u \leq A \\ \mathbf{0} & \text{if } u > A \end{cases}. \tag{B.6}$$

For $P(\mathbf{z})$ a Gaussian distribution with mean $\boldsymbol{\theta}$ and covariance $\boldsymbol{\Psi}$, we have

$$\log \frac{P(\mathbf{z}')}{P(\mathbf{z}_{t-1})} = -\frac{1}{2}(\boldsymbol{\Gamma}\mathbf{r}_{t-1} + \boldsymbol{\Gamma}\mathbf{e}_j - \boldsymbol{\theta})^\top \boldsymbol{\Psi}^{-1}(\boldsymbol{\Gamma}\mathbf{r}_{t-1} + \boldsymbol{\Gamma}\mathbf{e}_j - \boldsymbol{\theta})$$

$$+ \frac{1}{2}(\boldsymbol{\Gamma}\mathbf{r}_{t-1} - \boldsymbol{\theta})^\top \boldsymbol{\Psi}^{-1}(\boldsymbol{\Gamma}\mathbf{r}_{t-1} - \boldsymbol{\theta}) \tag{B.7}$$

$$= -\mathbf{e}_j^\top \boldsymbol{\Gamma}^\top \boldsymbol{\Psi}^{-1}(\boldsymbol{\Gamma}\mathbf{r}_{t-1} - \boldsymbol{\theta}) - \frac{1}{2}\mathbf{e}_j^\top \boldsymbol{\Gamma}^\top \boldsymbol{\Psi}^{-1}\boldsymbol{\Gamma}\mathbf{e}_j. \tag{B.8}$$

Defining the matrix

$$\boldsymbol{\Omega} \equiv \boldsymbol{\Gamma}^\top \boldsymbol{\Psi}^{-1}\boldsymbol{\Gamma}, \tag{B.9}$$

we interpret the first term in the log-odds ratio as a membrane potential,

$$\mathbf{V}_{t-1} = -\boldsymbol{\Omega}\mathbf{r}_{t-1} + \boldsymbol{\Gamma}^\top \boldsymbol{\Psi}^{-1}\boldsymbol{\theta}, \tag{B.10}$$

and the second as a threshold

$$T_j = \frac{1}{2}\Omega_{jj}, \tag{B.11}$$

such that the acceptance ratio is

$$A = \min\left\{1, \exp\left(V_{t-1,j} - T_j\right)\right\}. \tag{B.12}$$

With this definition, the membrane voltage simply integrates the spike train:

$$\mathbf{V}_t - \mathbf{V}_{t-1} = -\boldsymbol{\Omega}\mathbf{o}_t. \tag{B.13}$$

We now observe that, if $n_n > n_p$, it is possible that a spike may not contribute to the parameter estimate. Concretely, we say that a spike, or the corresponding neuron, is *irrelevant* if it is annihilated by $\boldsymbol{\Gamma}$, i.e., if $\boldsymbol{\Gamma}\mathbf{e}_j = \mathbf{0}$. Irrelevant spike proposals are always accepted with probability one, as we have $\mathbf{z}' = \mathbf{z}_{t-1}$. Moreover, the membrane voltage is not changed by the emission of irrelevant spikes. For these reasons, we are free to re-define the population rate $\mathbf{r}_t$ to exclude such spikes. Therefore, a timestep with an irrelevant spike is equivalent to not updating the network at all, and we could choose to re-define the network such that only relevant neurons are included. However, though there will exist some non-trivial set of vectors that are annihilated by $\boldsymbol{\Gamma}$ if $n_n > n_p$, the situation in which this null space is axis-aligned (thus implying the existence of irrelevant neurons) is not generic.

We note that appropriate initialization of the membrane potential (for the desired mean) is important, as otherwise some bias will be introduced. Thus, as written, this model cannot easily accommodate a time-varying mean signal $\boldsymbol{\theta}_t$. This shortcoming could obviously be addressed by taking

$$\mathbf{V}_t = -\boldsymbol{\Omega}\mathbf{r}_t + \boldsymbol{\Gamma}^\top \boldsymbol{\Psi}^{-1}\boldsymbol{\theta}_t, \tag{B.14}$$

which yields voltage dynamics

$$\mathbf{V}_t - \mathbf{V}_{t-1} = -\boldsymbol{\Omega}\mathbf{o}_t + \boldsymbol{\Gamma}^\top \boldsymbol{\Psi}^{-1}(\boldsymbol{\theta}_t - \boldsymbol{\theta}_{t-1}). \tag{B.15}$$

Alternatively, one could also take

$$\mathbf{V}_t = -\boldsymbol{\Omega}\mathbf{r}_t \tag{B.16}$$

and re-define the threshold for the $j$-th neuron to be time-varying:

$$(\boldsymbol{\Gamma}^\top \boldsymbol{\Psi}^{-1}\boldsymbol{\theta}_t)_j - \frac{1}{2}\Omega_{jj}. \tag{B.17}$$

The former of these approaches is reasonable from a biological perspective, as the new term $\boldsymbol{\Gamma}^\top \boldsymbol{\Psi}^{-1}(\boldsymbol{\theta}_t - \boldsymbol{\theta}_{t-1})$ in the voltage dynamics has the interpretation of a signal $\boldsymbol{\theta}_t - \boldsymbol{\theta}_{t-1}$ fed through an input weight matrix $\boldsymbol{\Gamma}^\top \boldsymbol{\Psi}^{-1}$. For this sampling procedure to work, the mean should be slowly-varying.

## B.2   Relaxing the assumption of perfect integration

We now relax the assumption of the perfect integration, and assume only that $\eta \ll 1$. With the same proposal distribution as before, we take the acceptance ratio of the accept-reject step to be

$$A = \min\left\{1, \frac{P[(1-\eta)\boldsymbol{\Gamma}\mathbf{r}_{t-1} + \boldsymbol{\Gamma}\mathbf{e}_j]}{P[(1-\eta)\boldsymbol{\Gamma}\mathbf{r}_{t-1}]}\right\}. \tag{B.18}$$

As noted in the main text, this choice implements a sort of look-ahead step. Moreover, if we took the acceptance ratio to depend on the likelihood of the proposed state with decay, $P[(1-\eta)\boldsymbol{\Gamma}\mathbf{r}_{t-1} + \boldsymbol{\Gamma}\mathbf{e}_j]$, relative to the likelihood $P[\boldsymbol{\Gamma}\mathbf{r}_{t-1}]$ of the current state (rather than the next state with decay but without the proposed spike), the resulting log-odds ratio would include terms of order $\eta$ that are quadratic in the rate $\mathbf{r}_t$.

With the choices above, the log-odds ratio is

$$\log \frac{P[(1-\eta)\boldsymbol{\Gamma}\mathbf{r}_{t-1} + \boldsymbol{\Gamma}\mathbf{e}_j]}{P[(1-\eta)\boldsymbol{\Gamma}\mathbf{r}_{t-1}]} = -\mathbf{e}_j^\top \boldsymbol{\Gamma}^\top \boldsymbol{\Psi}^{-1}[(1-\eta)\boldsymbol{\Gamma}\mathbf{r}_{t-1} - \boldsymbol{\theta}] - \frac{1}{2}\mathbf{e}_j^\top \boldsymbol{\Gamma}^\top \boldsymbol{\Psi}^{-1}\boldsymbol{\Gamma}\mathbf{e}_j \tag{B.19}$$

which, like in the perfect integrator model, is linear in the rate. As in the perfect integrator model, we define the recurrent weight matrix

$$\boldsymbol{\Omega} \equiv \boldsymbol{\Gamma}^\top \boldsymbol{\Psi}^{-1}\boldsymbol{\Gamma}, \tag{B.20}$$

and interpret the first term in the log-odds ratio as a membrane potential,

$$\mathbf{V}_{t-1} = -(1-\eta)\boldsymbol{\Omega}\mathbf{r}_{t-1} + \boldsymbol{\Gamma}^\top \boldsymbol{\Psi}^{-1}\boldsymbol{\theta}, \tag{B.21}$$

and the second as a threshold

$$T_j = \frac{1}{2}\Omega_{jj}, \tag{B.22}$$

such that the acceptance ratio is

$$A = \min\left\{1, \exp\left(V_{t-1,j} - T_j\right)\right\}. \tag{B.23}$$

With this definition, the membrane voltage evolves as

$$\mathbf{V}_t - (1-\eta)\mathbf{V}_{t-1} = -(1-\eta)\boldsymbol{\Omega}\mathbf{o}_t + \eta\boldsymbol{\Gamma}^\top \boldsymbol{\Psi}^{-1}\boldsymbol{\theta}. \tag{B.24}$$

Therefore, this model differs from the perfect integrator model of §B.1 only in the voltage dynamics; the perfect integrator is recovered exactly if we set $\eta = 0$. As in the perfect integrator model, the natural generalization of these leaky dynamics to time-varying mean signal $\boldsymbol{\theta}_t$ is to take

$$\mathbf{V}_t = -(1-\eta)\boldsymbol{\Omega}\mathbf{r}_t + \boldsymbol{\Gamma}^\top \boldsymbol{\Psi}^{-1}\boldsymbol{\theta}_t, \tag{B.25}$$

which leads to the dynamics

$$\mathbf{V}_t - (1-\eta)\mathbf{V}_{t-1} = -(1-\eta)\boldsymbol{\Omega}\mathbf{o}_t + \boldsymbol{\Gamma}^\top \boldsymbol{\Psi}^{-1}[\boldsymbol{\theta}_t - (1-\eta)\boldsymbol{\theta}_{t-1}]. \tag{B.26}$$

This will, of course, not be an exact sampler unless the mean is constant. If the mean is slowly-varying, however, it should be a good approximate sampler. We remark that we can re-write the membrane voltage as

$$\mathbf{V}_t = \boldsymbol{\Gamma}^\top \boldsymbol{\Psi}^{-1}[\boldsymbol{\theta}_t - (1-\eta)\mathbf{z}_t]. \tag{B.27}$$

## B.3 Adding an elastic net prior on the rates

We now consider the effect of adding an elastic net prior on the rates, as was considered in the original work of Boerlin et al. [3]:

$$P_{\text{e-net}}(\mathbf{r}) \propto \exp\left(-\alpha\|\mathbf{r}\|_1 - \frac{1}{2}\lambda\|\mathbf{r}\|_2^2\right). \tag{B.28}$$

Without loss of generality, we consider the case in which a decay term is included, as we can then recover the perfect integrator by setting $\eta = 0$. Again defining the acceptance ratio in terms of a comparison against the rate with decay but without the proposed spike, the addition of the prior adds

$$\log\frac{P_{\text{e-net}}[(1-\eta)\mathbf{r}_{t-1} + \mathbf{e}_j]}{P_{\text{e-net}}[(1-\eta)\mathbf{r}_{t-1}]} = -\alpha\|(1-\eta)\mathbf{r}_{t-1} + \mathbf{e}_j\|_1 - \frac{1}{2}\lambda\|(1-\eta)\mathbf{r}_{t-1} + \mathbf{e}_j\|_2^2$$

$$- \left(-\alpha\|(1-\eta)\mathbf{r}_{t-1}\|_1 - \frac{1}{2}\lambda\|(1-\eta)\mathbf{r}_{t-1}\|_2^2\right) \tag{B.29}$$

$$= -\alpha - \lambda(1-\eta)\mathbf{e}_j^\top \mathbf{r}_{t-1} - \frac{1}{2}\lambda \tag{B.30}$$

to the log-odds ratio. This yields a modified recurrent weight matrix

$$\mathbf{\Omega} \equiv \mathbf{\Gamma}^\top \mathbf{\Psi}^{-1}\mathbf{\Gamma} + \lambda\mathbf{I}_{n_n} \tag{B.31}$$

and a modified threshold

$$T_j = \frac{1}{2}\Omega_{jj} + \alpha, \tag{B.32}$$

but the expression for the membrane voltage in terms of these parameters is identical in functional form:

$$\mathbf{V}_t = -(1-\eta)\mathbf{\Omega}\mathbf{r}_t + \mathbf{\Gamma}^\top \mathbf{\Psi}^{-1}\boldsymbol{\theta}_t. \tag{B.33}$$

Therefore, adding the elastic net prior changes the definitions of the weight matrix that maps rates to voltages and of the threshold, but not the overall form of the result, hence it does not introduce any new conceptual difficulties. We remark that we could include the constant factor $\alpha$ in the membrane voltage as we do in our implementation of EBNs (see Appendix C) rather than in the threshold, which would give it a somewhat different biological interpretation but would have no algorithmic effect.

## B.4 The continuous-time limit

In this subsection, we consider the continuous-time limit of the models introduced above. This limit corresponds to taking the limit in which spike proposals are made infinitely often, and regarding the dynamics written down previously as a forward Euler discretization of an underlying continuous-time system. For clarity, we write the discrete timesteps, denoted in previous sections simply by $t$, as $t_d$ here, and reserve the unsubscripted symbol $t$ for the continuum variable. For a timestep $\Delta$, we let $t = \Delta t_d$, and let the discrete-time decay rate be $\eta = \Delta/\tau_m$ for a 'membrane' time constant $\tau_m$. We may then write the discretized rate dynamics (B.2) as

$$\tau_m \frac{\mathbf{r}(\Delta t_d) - \mathbf{r}(\Delta t_d - \Delta)}{\Delta} + \mathbf{r}(\Delta t_d - \Delta) = \tau_m \frac{1}{\Delta}\mathbf{o}_{t_d} \tag{B.34}$$

which, taking $\Delta \downarrow 0$, of course yields the familiar continuous-time dynamics

$$\tau_m \frac{d\mathbf{r}(t)}{dt} + \mathbf{r}(t) = \tau_m \mathbf{o}(t). \tag{B.35}$$

In continuous time, the spike train $\mathbf{o}(t)$ is now composed of Dirac delta functions, as the discretized spikes are rectangular pulses of width $\Delta$ in time and height $1/\Delta$. We next consider the voltage dynamics of the leaky integrator for a varying mean signal (B.26), which may similarly be re-written as

$$\tau_m \frac{\mathbf{V}(\Delta t_d) - \mathbf{V}(\Delta t_d - \Delta)}{\Delta} + \mathbf{V}(\Delta t_d - \Delta)$$

$$= -\mathbf{\Omega}\left(\tau_m \frac{1}{\Delta}\mathbf{o}_{t_d} - \mathbf{o}_{t_d}\right) + \mathbf{\Gamma}^\top \mathbf{\Psi}^{-1}\left(\tau_m \frac{\boldsymbol{\theta}(\Delta t_d) - \boldsymbol{\theta}(\Delta t_d - \Delta)}{\Delta} + \boldsymbol{\theta}(\Delta t_d - \Delta)\right). \tag{B.36}$$

In the continuum limit, we retain only the contribution of the first of the two terms involving the discrete-time spike train, as the other yields pulses of width $\Delta$ and height unity, which yield a negligible contribution to the integral. Thus, we have the dynamics

$$\tau_m \frac{d\mathbf{V}(t)}{dt} + \mathbf{V}(t) = -\tau_m \mathbf{\Omega}\mathbf{o}(t) + \mathbf{\Gamma}^\top \mathbf{\Psi}^{-1} \left( \tau_m \frac{d\boldsymbol{\theta}(t)}{dt} + \boldsymbol{\theta}(t) \right). \tag{B.37}$$

From these dynamics, one could then recover the continuum limit of the perfect integrator dynamics by taking $\tau_m \to \infty$. In this limit, the rate will decay by only a infinitesimal amount between a rejected spike proposal and the next proposal, meaning that the error incurred by neglecting the asymmetry in the proposal distribution due to the decay should be negligible.

## B.5 Analyzing the strongly-correlated limit of sampling from an equicorrelated Gaussian

Here, we consider how these models behave when sampling from an equicorrelated Gaussian distribution, i.e., a distribution with covariance matrix

$$\mathbf{\Psi} = (1 - \rho)\mathbf{I}_{n_p} + \rho \mathbf{1}_{n_p} \mathbf{1}_{n_p}^\top \tag{B.38}$$

for correlation coefficient $\rho \in (-1, +1)$. We are particularly interested in the strongly-correlated limit $\rho \uparrow 1$. As we will consider the case in which the marginal variance does not scale with the correlation, our choice of unit marginal variance is made without loss of generality. For this covariance matrix, the Sherman-Morrison formula yields [4]

$$\mathbf{\Psi}^{-1} = \frac{1}{1 - \rho} \left( \mathbf{I}_{n_p} - \frac{\rho}{1 + (n_p - 1)\rho} \mathbf{1}_{n_p} \mathbf{1}_{n_p}^\top \right). \tag{B.39}$$

We first consider the case in which the mean signal $\boldsymbol{\theta}$ is identically equal to zero. We argue that, for choices of $\mathbf{\Gamma}$ that are in some sense sufficiently naïve, the probability that relevant spikes are emitted should tend to zero as $\rho \uparrow 1$. This corresponds to showing that $V_{t,j} - T_j \to -\infty$ as $\rho \uparrow 1$ for all indices $j$ corresponding to relevant neurons. For $\boldsymbol{\theta} = \mathbf{0}$, we have

$$V_{t,j} - T_j = -\mathbf{e}_j^\top \mathbf{\Gamma}^\top \mathbf{\Psi}^{-1} \mathbf{\Gamma} \left( (1 - \eta)\mathbf{r}_t + \frac{1}{2}\mathbf{e}_j \right). \tag{B.40}$$

Consider the first timestep, with $\mathbf{r}_0 = \mathbf{0}$, for which we have

$$V_{0,j} - T_j = -\frac{1}{2}\mathbf{e}_j^\top \mathbf{\Gamma}^\top \mathbf{\Psi}^{-1} \mathbf{\Gamma}\mathbf{e}_j \tag{B.41}$$

$$= -\frac{1}{2(1 - \rho)} \left( (\mathbf{\Gamma}^\top \mathbf{\Gamma})_{jj} - \frac{\rho}{1 + (n_p - 1)\rho} (\mathbf{1}_{n_p}^\top \mathbf{\Gamma}\mathbf{e}_j)^2 \right). \tag{B.42}$$

Near $\rho = 1$, we then have the expansion

$$V_{0,j} - T_j = -\frac{1}{2(1 - \rho)} \left( (\mathbf{\Gamma}^\top \mathbf{\Gamma})_{jj} - \frac{1}{n_p} (\mathbf{1}_{n_p}^\top \mathbf{\Gamma}\mathbf{e}_j)^2 \right) + \mathcal{O}(1), \tag{B.43}$$

under the assumption that $\mathbf{\Gamma}$ is an $\mathcal{O}(1)$ function of $\rho$ in this region, and thus cannot introduce additional possible divergences. For relevant spikes, we have the strict inequality $(\mathbf{\Gamma}^\top \mathbf{\Gamma})_{jj} > 0$. Moreover, by the Cauchy-Schwarz inequality, we have $(\mathbf{1}_{n_p}^\top \mathbf{\Gamma}\mathbf{e}_j)^2 \leq n_p (\mathbf{\Gamma}^\top \mathbf{\Gamma})_{jj}$, with equality if and only if $\mathbf{\Gamma}\mathbf{e}_j \propto \mathbf{1}_{n_p}$. If $\mathbf{\Gamma}\mathbf{e}_j \propto \mathbf{1}_{n_p}$, then spikes in neuron $j$ affect the readout precisely only along the common mode, and $V_{0,j} - T_j$ does not diverge as $\rho \uparrow 1$. However, this case is quite fine-tuned. For generic $\mathbf{\Gamma}$ not satisfying this alignment condition, we have the strict inequality

$$(\mathbf{\Gamma}^\top \mathbf{\Gamma})_{jj} - \frac{1}{n_p} (\mathbf{1}_{n_p}^\top \mathbf{\Gamma}\mathbf{e}_j)^2 > 0 \tag{B.44}$$

for all relevant neurons. Then, if $(\mathbf{\Gamma}^\top \mathbf{\Gamma})_{jj}$ vanishes no more rapidly than $1 - \rho$ as $\rho \uparrow 1$—i.e., if $(\mathbf{\Gamma}^\top \mathbf{\Gamma})_{jj}/(1 - \rho) \to \infty$ as $\rho \uparrow 1$—we have that $V_{0,j} - T_j \to -\infty$ as $\rho \uparrow 1$. This holds, for instance, if $(\mathbf{\Gamma}^\top \mathbf{\Gamma})_{jj}$ is a constant function of $\rho$. Thus, under these conditions, the probability that a relevant spike is emitted at the first timestep vanishes as $\rho \uparrow 1$. Heuristically, this in turn implies that the (relevant component of the) rate at the second timestep will be zero with probability one. Therefore,

we may iterate this argument forward in time, showing that the probability of emission of relevant spikes should vanish in the limit $\rho \uparrow 1$. This argument will not be affected by the addition of an elastic net penalty unless the coefficients $\alpha$ and $\lambda$ are taken to diverge as $\rho$ is taken to unity, as the coefficients are strictly non-positive.

The situation is somewhat more complicated if the mean signal is not identically zero, in which case we have

$$V_{t,j} - T_j = -(1-\eta)\mathbf{e}_j^\top \mathbf{\Gamma}^\top \mathbf{\Psi}^{-1} \mathbf{\Gamma} \mathbf{r}_t - \frac{1}{2}(\mathbf{\Gamma}^\top \mathbf{\Psi}^{-1} \mathbf{\Gamma})_{jj} + \mathbf{\Gamma}^\top \mathbf{\Psi}^{-1} \boldsymbol{\theta}_t. \tag{B.45}$$

Following our previous analysis, at the first timestep we have

$$V_{0,j} - T_j = -\frac{1}{2}\mathbf{e}_j \mathbf{\Gamma}^\top \mathbf{\Psi}^{-1} \mathbf{\Gamma} \mathbf{e}_j + \mathbf{e}_j^\top \mathbf{\Gamma}^\top \mathbf{\Psi}^{-1} \boldsymbol{\theta}_t. \tag{B.46}$$

$$= -\frac{1}{2(1-\rho)}\left((\mathbf{\Gamma}^\top \mathbf{\Gamma})_{jj} - \frac{1}{n_p}(\mathbf{1}_{n_p}^\top \mathbf{\Gamma} \mathbf{e}_j)^2\right)$$

$$+ \frac{1}{1-\rho}\left(\mathbf{e}_j^\top \mathbf{\Gamma}^\top \boldsymbol{\theta}_t - \frac{1}{n_p}(\mathbf{1}_{n_p}^\top \mathbf{\Gamma} \mathbf{e}_j)(\mathbf{1}_{n_p}^\top \boldsymbol{\theta}_t)\right) + \mathcal{O}(1) \tag{B.47}$$

near $\rho = 1$. There are now two possible divergent terms, which can compete to change the sign of $V_{0,j} - T_j$ as $\rho \uparrow 1$. One case of interest is when $\boldsymbol{\theta}_t \propto \mathbf{1}_{n_p}$. Then,

$$\mathbf{e}_j^\top \mathbf{\Gamma}^\top \boldsymbol{\theta}_t - \frac{1}{n_p}(\mathbf{1}_{n_p}^\top \mathbf{\Gamma} \mathbf{e}_j)(\mathbf{1}_{n_p}^\top \boldsymbol{\theta}_t) = 0 \tag{B.48}$$

and we have

$$V_{0,j} - T_j = -\frac{1}{2(1-\rho)}\left((\mathbf{\Gamma}^\top \mathbf{\Gamma})_{jj} - \frac{1}{n_p}(\mathbf{1}_{n_p}^\top \mathbf{\Gamma} \mathbf{e}_j)^2\right) + \mathcal{O}(1), \tag{B.49}$$

as in the case when $\boldsymbol{\theta}$ was strictly zero, implying that $V_{0,j} - T_j \to -\infty$ under the abovementioned conditions on $\mathbf{\Gamma}$. Another illustrative case is $\boldsymbol{\theta}_t = \mathbf{\Gamma} \mathbf{e}_j$. With this fine-tuning,

$$V_{0,j} - T_j = \frac{1}{2(1-\rho)}\left((\mathbf{\Gamma}^\top \mathbf{\Gamma})_{jj} - \frac{1}{n_p}(\mathbf{1}_{n_p}^\top \mathbf{\Gamma} \mathbf{e}_j)^2\right) + \mathcal{O}(1), \tag{B.50}$$

hence we expect $V_{0,j} - T_j \to +\infty$ as $\rho \uparrow 1$ under the abovementioned constraints on $\mathbf{\Gamma}$. Thus, in this case, the spike probability should tend to one, showing that complications can arise in the case of a non-uniform mean signal.

## B.6 Analyzing the small- and large-variance limits

We now consider how these models behave in the limits in which the variance of the target distribution is small or large, or, nearly equivalently, the limits in which the scale of the readout matrix is large or small, respectively. In §2.3 of the main text, we argued that the EBN is recovered in the limit in which the variance of the target distribution tends to zero. Concretely, we let $\mathbf{\Psi} = \psi \tilde{\mathbf{\Psi}}$ for some fixed matrix $\tilde{\mathbf{\Psi}}$, and take the zero-variance limit $\psi \downarrow 0$. In terms of the renormalized voltage $\tilde{\mathbf{V}} \equiv \psi \mathbf{V}$, weight matrix $\tilde{\mathbf{\Omega}} = \psi \mathbf{\Omega}$, and threshold $\tilde{\mathbf{T}} = \psi \mathbf{T}$, the spike acceptance ratio is $A = \min\{1, \exp[(\tilde{V}_j(t) - \tilde{T}_j)/\psi]\}$. This tends to $A = \Theta(\tilde{V}_j(t) - \tilde{T}_j)$ as $\psi \downarrow 0$ assuming that the re-scaled variables remain order one, recovering the greedy spiking rule used in the EBN. If we instead take the large-variance limit $\psi \uparrow \infty$, the acceptance ratio tends to 1 under the assumption that $\tilde{V}_j(t) - \tilde{T}_j$ remains $\mathcal{O}(1)$. Then, each spike proposal is accepted with probability one, and the total firing rate of the population is $1/\Delta$ spikes per second for a timestep $\Delta$, meaning that the population-averaged rate is $1/(n_n\Delta)$ spikes per second.

Re-scaling the readout matrix $\mathbf{\Gamma}$ differs from re-scaling the target covariance matrix $\mathbf{\Psi}$ because the target mean and recurrent spikes appear with the same power of $\mathbf{\Psi}$ (in particular, $\mathbf{\Psi}^{-1}$) but different powers of $\mathbf{\Gamma}$ in the voltage dynamics, scaling as $\mathcal{O}(\mathbf{\Gamma})$ and $\mathcal{O}(\mathbf{\Gamma}^2)$, respectively. Thus, in the large-$\mathbf{\Gamma}$ limit in discrete time, the contribution of the target mean should be negligible relative to that of the recurrent spiking input, while the opposite should hold in the small-$\mathbf{\Gamma}$ limit. However, these different scalings should not affect the limiting behavior of the acceptance ratio. In Figure B.1, we probe how sweeping the scale of $\mathbf{\Gamma}$ over five orders of magnitude affects sampling from a Gaussian distribution of fixed variance. We show that the population-averaged spike rate tends to $1/(n_n\Delta)$ spikes per second in the small-$\mathbf{\Gamma}$ limit, while the spike rate tends to zero in the large-$\mathbf{\Gamma}$ limit.

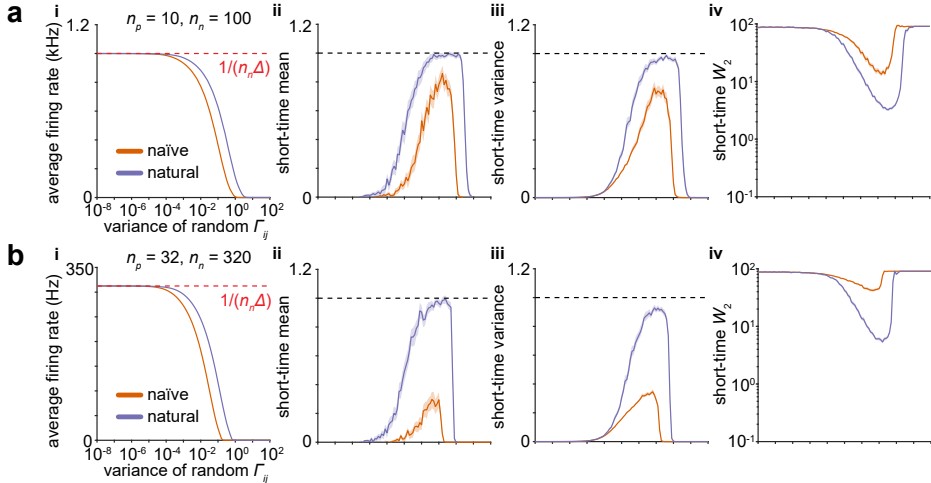

Figure B.1: Sensitivity of the Metropolis-Hastings sampler to the scale of the readout matrix. **a**. Sampling performance of a network of $n_n = 100$ neurons sampling from a $n_p = 10$-dimensional equicorrelated Gaussian with $\rho = 0.75$ depends on the variance of the elements of the random readout matrix $\boldsymbol{\Gamma}$. The stimulus setup is as in Figure 3. **a.i**. At large variance, spiking is suppressed in networks with naïve or natural geometry. At small variance, spikes are accepted with probability one, and the population-averaged firing rate tends to $1/(n_n\Delta)$, as indicated by the red dashed line. **a.ii**. Dimension-averaged estimate of the mean signal over the entire stimulus interval. **a.iii**. As in **ii**, but for the variance. **a.iv**. As in **ii**, but for the 2-Wasserstein distance. **b**. As in **a**, but for a network of $n_n = 320$ neurons sampling from an $n_p = 32$-dimensional Gaussian. Shaded error patches show 95% confidence intervals computed via bootstrapping over 100 realizations in all panels; see Appendix E for further details.

## B.7 Alternative spike proposal distributions and membrane voltage bounds

In this appendix, we have considered the simplest possible spike proposal distribution: at each timestep, we choose one neuron uniformly at random as a candidate. With this choice, obviously, only one neuron can spike at each timestep, which parallels the spiking rule used in the EBN: only the neuron with the maximum membrane voltage among the entire population is allowed to spike [3, 5]. With a discretization timestep $\Delta$, the maximum total spike rate is then $1/\Delta$. In the EBN literature, previous analysis has shown that this constraint is vital to avoid pathological spiking patterns [3, 5–8], and introduced alternatives such as Poisson spiking rules [8] or hand-tuned refractory periods [9]. We remark that, in these models, the membrane voltage is always strictly bounded from above, which does not hold in our setting. As reset is achieved only through spike emission, and is strictly speaking a decrement of the voltage rather than a true reset, a neuron can exceed the threshold voltage if it is not chosen as a candidate to spike.

## B.8 Sampling from non-Gaussian exponential families

Though our main focus is on Gaussian target distributions, in this appendix we briefly discuss the possibility of constructing a probabilistic spiking sampler for other exponential families. For a distribution with density

$$P(\mathbf{z}) = \exp[-U(\mathbf{z})] \tag{B.51}$$

for an energy function $U$, the acceptance ratio (B.18) becomes

$$A = \min\left\{1, \exp\left(U[(1-\eta)\boldsymbol{\Gamma}\mathbf{r}_{t-1}] - U[(1-\eta)\boldsymbol{\Gamma}\mathbf{r}_{t-1} + \boldsymbol{\Gamma}\mathbf{e}_j]\right)\right\}. \tag{B.52}$$

In analogy to the Gaussian case, one could then define the difference of energies to be the difference of the membrane voltage and the threshold. However, the resulting membrane voltage would be a nonlinear function of the firing rate, and would in general evolve according to non-linear dynamics.

Most directly, assuming $\|\mathbf{\Gamma}\mathbf{e}_j\|$ to be small and $U$ to be not too quickly varying, one could make the second-order approximation

$$U[(1-\eta)\mathbf{\Gamma}\mathbf{r}_{t-1}] - U[(1-\eta)\mathbf{\Gamma}\mathbf{r}_{t-1} + \mathbf{\Gamma}\mathbf{e}_j] \approx -\nabla U[(1-\eta)\mathbf{\Gamma}\mathbf{r}_{t-1}]^\top \mathbf{\Gamma}\mathbf{e}_j$$
$$-\frac{1}{2}\mathbf{e}_j^\top \mathbf{\Gamma}^\top \nabla^2 U[(1-\eta)\mathbf{\Gamma}\mathbf{r}_{t-1}]\mathbf{\Gamma}\mathbf{e}_j, \quad \text{(B.53)}$$

and define the membrane voltage and threshold as

$$\mathbf{V}_{t-1} = -\mathbf{\Gamma}^\top \nabla U[(1-\eta)\mathbf{\Gamma}\mathbf{r}_{t-1}] \quad \text{(B.54)}$$

and

$$T_{j,t-1} = \frac{1}{2}(\mathbf{\Gamma}^\top \nabla^2 U[(1-\eta)\mathbf{\Gamma}\mathbf{r}_{t-1}]\mathbf{\Gamma})_{jj}, \quad \text{(B.55)}$$

respectively. With this choice, the threshold would be state-dependent, but one could incorporate its state-dependence into a re-defined voltage.

## C  Sampling in efficient balanced networks

### C.1  Encoding a dynamical system in efficient balanced networks

In this section, we provide a pedagogical derivation of the efficient balanced network [3, 5] using the notation we use throughout the paper. The goal of the network is to encode an estimate $\mathbf{z}(t)$ of a signal $\boldsymbol{\theta}$ (a vector of size $n_p \times 1$ in a population of $n_n$ spiking neurons. The estimate is obtained by reading out a low pass version of the population spiking activity:

$$\mathbf{z}(t) = \mathbf{\Gamma}r(t) \quad \text{(C.1)}$$

where $\mathbf{\Gamma}$ is the $n_p \times n_n$ readout matrix and $r(t)$ is the low-pass filter spike history:

$$\frac{dr_i(t)}{dt} = -\frac{1}{\tau_m}r_i(t) + o_i(t) \quad \text{(C.2)}$$

where $\tau_m$ is the time constant of the readout neurons and $o_i$ is the spike train of neuron $i$, $o_i = 1$ if the neuron spiked at $t$ and $o_i = 0$ otherwise.

The goal of the network is to minimize the squared error between the signal and the estimate with an elastic net prior on the firing rate in order to find a good solution while keeping the population spiking activity relatively low:

$$\mathcal{L} = (\boldsymbol{\theta} - \mathbf{z})^\top (\boldsymbol{\theta} - \mathbf{z}) + 2\alpha \mathbf{r} + \lambda \mathbf{r}^\top \mathbf{r} \quad \text{(C.3)}$$

In the standard implementation of efficient balanced networks, neurons use a greedy spiking rule. A neuron should fire if emitting a spike will lower the loss function, i.e., if

$$\mathcal{L}(o_i^t = 0) > \mathcal{L}(o_i^t = 1). \quad \text{(C.4)}$$

Using the loss function and the definition of the estimate ($\mathbf{z}(t) = \mathbf{\Gamma}r(t)$), we can rewrite the spiking rule for neuron $i$ as:

$$(\boldsymbol{\theta} - \mathbf{z})^\top (\boldsymbol{\theta} - \mathbf{z}) + 2\alpha \sum_j r_j + \lambda \sum_j r_j^2$$
$$> (\boldsymbol{\theta} - \mathbf{z} - \mathbf{\Gamma}_i)^\top (\boldsymbol{\theta} - \mathbf{z} - \mathbf{\Gamma}_i) + 2\alpha(1 + \sum_j r_j) + \lambda((1+r_i)^2 + \sum_{j \neq i} r_j^2). \quad \text{(C.5)}$$

Removing terms that appear on both sides, we get

$$0 > (\boldsymbol{\theta} - \mathbf{z})^\top (-\mathbf{\Gamma}_i) + \mathbf{\Gamma}_i^\top (\boldsymbol{\theta} - \mathbf{z} - \mathbf{\Gamma}_i) + 2\alpha + 2\lambda r_i + \lambda, \quad \text{(C.6)}$$

which we can further simplify to

$$\mathbf{\Gamma}_i^\top (\boldsymbol{\theta} - \mathbf{z}) - \alpha - \lambda r_i > \frac{1}{2}(\mathbf{\Gamma}_i^\top \mathbf{\Gamma} + \lambda). \quad \text{(C.7)}$$

The term on the left hand can be interpreted as the voltage potential of neuron $i$:

$$V_i = \mathbf{\Gamma}_i^\top (\boldsymbol{\theta} - \mathbf{z}) - \alpha - \lambda r_i, \tag{C.8}$$

while the term on the right hand size can be interpreted as the firing threshold[3]:

$$T_i = \frac{1}{2}(\lambda + \mathbf{\Gamma}_i^\top \mathbf{\Gamma}_i). \tag{C.9}$$

If the voltage potential exceeds this threshold, then the neuron will fire and lower its voltage potential back below threshold. Using (C.8) we can express the dynamics of the voltage potential as a function of the dynamics of $\boldsymbol{\theta}$ and its estimate $\mathbf{z}$:

$$\frac{dV_i(t)}{dt} = \mathbf{\Gamma}_i^\top \left( \frac{d\boldsymbol{\theta}}{dt} - \frac{d\mathbf{z}}{dt} \right) - \lambda \frac{dr_i}{dt}. \tag{C.10}$$

We can rewrite this equation to obtain an expression for the membrane dynamics as a function of the signal, the firing rates, and the spike trains of the neurons. By adding and subtracting $\frac{1}{\tau_m}\mathbf{\Gamma}_i^\top \boldsymbol{\theta} - \frac{1}{\tau_m}\alpha$ and noting that $\mathbf{\Gamma}_i^\top \mathbf{\Gamma} \mathbf{r} + \lambda r_i - \mathbf{\Gamma}_i^\top \boldsymbol{\theta} + \alpha = -V_i$, we further simplify the expression to obtain membrane dynamics as a function of the membrane potential, the effect of a new spike on the circuit and the encoded dynamical system:

$$\frac{dV_i}{dt} = \mathbf{\Gamma}_i^\top \left[ \frac{d\boldsymbol{\theta}}{dt} - \mathbf{\Gamma} \left( -\frac{1}{\tau_m}\mathbf{r} + \mathbf{o} \right) \right] + \frac{1}{\tau_m}\lambda r_i - \lambda o_i \tag{C.11}$$

$$= -(\mathbf{\Gamma}_i^\top \mathbf{\Gamma}_i + \lambda)o_i + \mathbf{\Gamma}_i^\top \frac{d\boldsymbol{\theta}}{dt} + \frac{1}{\tau_m}\mathbf{\Gamma}_i^\top \mathbf{\Gamma}\mathbf{r} + \frac{1}{\tau_m}\lambda r_i$$

$$- \frac{1}{\tau_m}\mathbf{\Gamma}_i^\top \boldsymbol{\theta} + \frac{1}{\tau_m}\alpha + \frac{1}{\tau_m}\mathbf{\Gamma}_i^\top \boldsymbol{\theta} - \frac{1}{\tau_m}\alpha \tag{C.12}$$

$$= -\frac{1}{\tau_m}V_i - \frac{1}{\tau_m}\alpha - (\mathbf{\Gamma}_i^\top \mathbf{\Gamma}_i + \lambda)o_i + \mathbf{\Gamma}_i^\top \left( \frac{d\boldsymbol{\theta}}{dt} + \frac{1}{\tau_m}\boldsymbol{\theta} \right). \tag{C.13}$$

We therefore obtain the membrane dynamics for the efficient balanced spiking network proposed in [3, 5]. We can rewrite the dynamics in vector form:

$$\frac{d\mathbf{V}}{dt} = -\frac{1}{\tau_m}\mathbf{V} - \frac{1}{\tau_m}\alpha - \mathbf{\Omega}\mathbf{o} + \mathbf{\Gamma}^\top \left( \frac{d\boldsymbol{\theta}}{dt} + \frac{1}{\tau_m}\boldsymbol{\theta} \right), \quad \text{with} \quad \mathbf{\Omega} = (\mathbf{\Gamma}^\top \mathbf{\Gamma} + \lambda \mathbf{I}_{n_n}). \tag{C.14}$$

Using this scheme one can encode of variety of dynamical systems, including Langevin dynamics [5–7].

### C.2  Sampling in efficient balanced networks using naïve Langevin dynamics

Using the scheme presented in §C.1 above, Savin and Denève [5] proposed to implement a dynamical system corresponding to the naïve Langevin dynamics of a multivariate normal distribution. We use a linear Gaussian model similarly to several studies of neuroscience inspired sampling-based networks [5, 10]. These networks estimate the posterior probability of hidden sources ($\boldsymbol{\theta}$) given sensory inputs $\mathbf{x}$ corrupted by Gaussian noise, $p(\mathbf{x}|\boldsymbol{\theta}) = \mathcal{N}(\mathbf{x}; \mathbf{A}\boldsymbol{\theta}, \sigma_n^2 \mathcal{I})$, and prior expectations on the values of the hidden sources $p(\boldsymbol{\theta}) = \mathcal{N}(\boldsymbol{\theta}; 0, \mathbf{C})$. The mean $\boldsymbol{\mu}$ and covariance $\mathbf{\Sigma}$ of the posterior probability of the features given the input, $p(\boldsymbol{\theta}|\mathbf{x}) \propto p(\mathbf{x}|\boldsymbol{\theta})p(\boldsymbol{\theta})$, are $\boldsymbol{\mu} = \frac{1}{\sigma_n^2}\mathbf{\Sigma}\mathbf{A}^\top \mathbf{x}$ and $\mathbf{\Sigma} = \left( \mathbf{C} + \frac{1}{\sigma_n^2}\mathbf{A}^\top \mathbf{A} \right)^{-1}$ respectively. Up to an irrelevant constant offset, the corresponding energy function is $U(\boldsymbol{\theta}) = \frac{1}{2}(\boldsymbol{\theta} - \boldsymbol{\mu})^\top \mathbf{\Sigma}^{-1}(\boldsymbol{\theta} - \boldsymbol{\mu})$, and its gradient is $\nabla U(\boldsymbol{\theta}) = \mathbf{\Sigma}^{-1}(\boldsymbol{\theta} - \boldsymbol{\mu})$. We define $\tau_s$ as the timescale of the inference process and we set $\epsilon_t = \frac{1}{\tau_s}$. Then, we can write down the naïve Langevin dynamics

$$d\boldsymbol{\theta}(t) = \left( -\frac{1}{\tau_s}\mathbf{\Sigma}^{-1}\boldsymbol{\theta} + \frac{1}{\tau_s}\mathbf{A}^\top \mathbf{x} \right) dt + \sqrt{\frac{2}{\tau_s}}d\mathbf{W}(t) \tag{C.15}$$

---

[3]Here, we chose to include the regularizing term $\alpha$ as a fixed offset in the voltage potential but it can equivalently be included as an offset in the spiking threshold $T_i$, as discussed in Appendix B.3.

These dynamics can be approximated by an efficient balanced network by replacing in (C.14) $\boldsymbol{\theta}$ by $\mathbf{z}$. As discussed in [5], this approximation will introduce an acceptable error in most practical situations:

$$\frac{d\mathbf{V}}{dt} = -\frac{1}{\tau_m}\mathbf{V} - \frac{1}{\tau_m}\alpha - \boldsymbol{\Omega}\mathbf{o} + \boldsymbol{\Gamma}^\top\left(\frac{d\mathbf{z}}{dt} + \frac{1}{\tau_m}\mathbf{z}\right), \tag{C.16}$$

and we therefore obtain the membrane dynamics for the efficient balanced network proposed by Savin and Denève [5]:

$$d\mathbf{V}(t) = \left[-\frac{1}{\tau_m}\mathbf{V} - \frac{1}{\tau_m}\alpha - \boldsymbol{\Omega}\mathbf{o} + \frac{1}{\tau_s}\boldsymbol{\Gamma}^\top(\mathbf{A}^\top\mathbf{x} - \boldsymbol{\Sigma}^{-1}\boldsymbol{\Gamma}\mathbf{r}) + \frac{1}{\tau_m}\boldsymbol{\Gamma}^\top\boldsymbol{\Gamma}\mathbf{r}\right]dt + \boldsymbol{\Gamma}^\top\sqrt{\frac{2}{\tau_s}}d\mathbf{W}(t) \tag{C.17}$$

Note here that we have two timescales: the timescale of neuronal representations $\tau_m \sim 20$ ms, controlled by the biophysical properties of the neurons, and the timescale $\tau_s$ of the Langevin diffusion encoded by the network.

### C.3  Sampling in efficient balanced networks using the complete recipe for stochastic gradient MCMC

The model proposed by Savin and Denève [5] implements naïve Langevin dynamics, which are known to be slow in high dimensions. Instead, we can use the "complete recipe" for stochastic gradient MCMC [11] to write another sampler with the same equilibrium distribution but more favorable convergence properties. For any positive semi-definite matrix $\mathbf{D}$ and skew-symmetric matrix $\mathbf{S}$, the following dynamics will converge to $\mathcal{N}(\boldsymbol{\mu}, \boldsymbol{\Sigma})$ as their stationary distribution:

$$d\boldsymbol{\theta}(t) = \left[-\frac{1}{\tau_s}(\mathbf{D} + \mathbf{S})(\boldsymbol{\Sigma}^{-1}\boldsymbol{\theta} - \mathbf{A}^\top\mathbf{x})\right]dt + \mathbf{B}\sqrt{\frac{2}{\tau_s}}d\mathbf{W}(t) \tag{C.18}$$

with $\mathbf{B}\mathbf{B}^\top = \mathbf{D}$.

We can use (C.14) to encode this more general formulation into an efficient balanced network, yielding the following membrane dynamics:

$$d\mathbf{V} = \frac{1}{\tau_m}\left[-\mathbf{V} - \alpha - \tau_m\boldsymbol{\Omega}\mathbf{o} + \boldsymbol{\Gamma}^\top\left(\mathbf{I}_{n_p} - \frac{\tau_m}{\tau_s}(\mathbf{D} + \mathbf{S})\boldsymbol{\Sigma}^{-1}\right)\boldsymbol{\Gamma}\mathbf{r} + \frac{\tau_m}{\tau_s}\boldsymbol{\Gamma}^\top(\mathbf{D} + \mathbf{S})\mathbf{A}^\top\mathbf{x}\right]dt$$
$$+ \boldsymbol{\Gamma}^\top\mathbf{B}\sqrt{\frac{2}{\tau_s}}d\mathbf{W}. \tag{C.19}$$

Any choice of positive semi-definite matrix $\mathbf{D}$ leads to a valid sampler, but extensive work inspired by Amari's seminal work on natural gradient descent [12, 13] has shown that a principled choice is to take $\mathbf{D}$ to be the inverse of the Fisher information matrix $\mathbf{G}$ [11, 14]. For the multivariate Gaussian distribution, the Fisher information matrix is given by:

$$G_{ij} = -\mathbb{E}\left[\frac{\partial^2}{\partial\theta_i\partial\theta_j}\log P(\boldsymbol{\theta})\right] \tag{C.20}$$

$$= \mathbb{E}\left[\frac{\partial^2}{\partial\theta_i\partial\theta_j}\frac{1}{2}(\boldsymbol{\theta} - \boldsymbol{\mu})^\top\boldsymbol{\Sigma}^{-1}(\boldsymbol{\theta} - \boldsymbol{\mu})\right] \tag{C.21}$$

$$= \Sigma_{ij}^{-1} \tag{C.22}$$

We should therefore choose $\mathbf{D} = \mathbf{G}^{-1} = \boldsymbol{\Sigma}$. Note that for the multivariate Gaussian distribution, the Fisher information matrix is identical to the Hessian and is location independent - it does not depend on the value of $\boldsymbol{\theta}$. For more complex distributions, the Fisher information matrix might be difficult to compute and an approximation can be used as long as it is valid (positive semi-definite) within the complete recipe framework [11, 15, 16] and state dependent matrices can be corrected for using the term $\boldsymbol{\Phi}$ from the complete recipe in eq. (12) of the main text [11].

In this work, we have considered only hand-tuned or random choices for the matrices controlling the geometry. Previous work by Hennequin et al. [10], when framed within the complete recipe, proposes methods to find a skew-symmetric matrix $\mathbf{S}$ which accelerates the dynamics. Non-reversibility is indeed known to accelerate learning, but analysing networks with such dynamics is notoriously difficult [17, 18]. In contrast, approximations for the inverse Fisher information matrix are readily computable, even in non-Gaussian settings [15, 16].

## C.4 Sampling from non-Gaussian exponential families using efficient balanced networks

To sample from a non-Gaussian exponential family distribution with energy $U(\boldsymbol{\theta})$ using an EBN, we can simulate the "complete recipe" for a general exponential-family as stated in eq. (12) of the main text, which we reproduce below:

$$d\mathbf{z}(t) = -\frac{1}{\tau_s}\{[\mathbf{D}(\mathbf{z}) + \mathbf{S}(\mathbf{z})]\boldsymbol{\nabla}U(\mathbf{z}) + \boldsymbol{\Phi}(\mathbf{z})\}\, dt + \mathbf{B}\sqrt{\frac{2}{\tau_s}}\, d\mathbf{W}(t), \qquad (C.23)$$

where $\mathbf{B}\mathbf{B}^\top = \mathbf{D}$ [19]. Here, we have introduced an auxiliary time constant $\tau_s$, as in our previous discussion of the Gaussian case. Then, the EBN voltage dynamics (C.14) become

$$d\mathbf{V}(t) = \left[-\frac{1}{\tau_m}\mathbf{V} - \frac{1}{\tau_m}\boldsymbol{\alpha} - \boldsymbol{\Omega}\mathbf{o} + \boldsymbol{\Gamma}^\top\left(-\frac{1}{\tau_s}\{[\mathbf{D}(\mathbf{z}) + \mathbf{S}(\mathbf{z})]\boldsymbol{\nabla}U(\mathbf{z}) + \boldsymbol{\Phi}(\mathbf{z})\} + \frac{1}{\tau_m}\mathbf{z}\right)\right]dt$$

$$+ \boldsymbol{\Gamma}^\top\mathbf{B}\sqrt{\frac{2}{\tau_s}}\, d\mathbf{W}(t) \qquad (C.24)$$

where we have again made the approximation of the Langevin sampling trajectory by the approximate sampling trajectory $\mathbf{z} =$. It is easy to see that these dynamics will in general involve non-linear dependence on the population firing rate $\mathbf{r}$ through the readout $\mathbf{z} = \boldsymbol{\Gamma}\mathbf{r}$.

# D Natural gradient enables fast sampling in linear rate networks

In this section, we analyze how natural gradients enable fast sampling in rate networks designed to sample from zero-mean Gaussian distributions. Our starting point is the complete recipe for sampling from a zero-mean Gaussian distribution with covariance $\boldsymbol{\Sigma}$:

$$d\mathbf{z}(t) = -(\mathbf{D} + \mathbf{S})\boldsymbol{\Sigma}^{-1}\mathbf{z}\, dt + \sqrt{2\mathbf{D}}\, d\mathbf{W}(t) \qquad (D.1)$$

where $\mathbf{D}$ is a symmetric positive-semidefinite matrix and $\mathbf{S}$ is skew-symmetric. In previous work, Hennequin et al. [10] showed how these dynamics can be interpreted as a linear rate network, and demonstrated that careful choice of $\mathbf{S}$ can accelerate sample autocorrelation timescales in networks with $\mathbf{D} = \mathbf{I}_{n_p}$. Here, our objective is to show how the geometry of inference, set by $\mathbf{D}$, can accelerate sampling in these rate networks.

Our analysis is a straightforward application of the classic theory of Ornstein-Uhlenbeck processes [20–22]. Assuming for simplicity a deterministic initial condition $\mathbf{z}(0) = \mathbf{0}$, the solution to this stochastic differential equation is given by the Itô integral

$$\mathbf{z}(t) = \sqrt{2}\int_0^t e^{-(\mathbf{D}+\mathbf{S})\boldsymbol{\Sigma}^{-1}(t-s)}\mathbf{B}\, d\mathbf{W}(s), \qquad (D.2)$$

which has ensemble mean zero and ensemble covariance

$$\mathbf{C}(t,t') \equiv \mathbb{E}\mathbf{z}(t)\mathbf{z}(t')^\top = 2\int_0^{\min(t,t')} ds\, e^{-(\mathbf{D}+\mathbf{S})\boldsymbol{\Sigma}^{-1}(t-s)}\mathbf{D}e^{-\boldsymbol{\Sigma}^{-1}(\mathbf{D}-\mathbf{S})(t'-s)} \qquad (D.3)$$

The ensemble distribution of $\mathbf{z}(t)$ is of course Gaussian, with mean zero and covariance given by the equal-time covariance function $\mathbf{C}(t) \equiv \mathbf{C}(t,t)$.

We assume that $(\mathbf{D} + \mathbf{S})\boldsymbol{\Sigma}^{-1}$ is a non-defective matrix with all eigenvalues having positive real part, such that the process has a well-defined stationary state. By construction, the covariance matrix of the stationary state is precisely the target covariance matrix $\boldsymbol{\Sigma}$. In the stationary state, we have a simplified two-point function

$$\mathbf{C}_s(t - t') \equiv \mathbb{E}_s\mathbf{z}(t)\mathbf{z}(t')^\top = \begin{cases} e^{-(\mathbf{D}+\mathbf{S})\boldsymbol{\Sigma}^{-1}(t-t')}\boldsymbol{\Sigma}, & t > t' \\ e^{-(\mathbf{D}-\mathbf{S})\boldsymbol{\Sigma}^{-1}(t'-t)}\boldsymbol{\Sigma} & t < t'. \end{cases} \qquad (D.4)$$

where we have observed that

$$\boldsymbol{\Sigma}e^{-\boldsymbol{\Sigma}^{-1}(\mathbf{D}-\mathbf{S})(t'-t)} = \sum_{j=0}^\infty \frac{(t-t')^j}{j!}\boldsymbol{\Sigma}[\boldsymbol{\Sigma}^{-1}(\mathbf{D}-\mathbf{S})]^j \qquad (D.5)$$

$$= \sum_{j=0}^\infty \frac{(t-t')^j}{j!}[(\mathbf{D}-\mathbf{S})\boldsymbol{\Sigma}^{-1}]^j\boldsymbol{\Sigma} \qquad (D.6)$$

$$= e^{-(\mathbf{D}-\mathbf{S})\boldsymbol{\Sigma}^{-1}(t'-t)}\boldsymbol{\Sigma} \qquad (D.7)$$

One standard case in which the integral defining the non-stationary covariance function can be evaluated explicitly is if $\mathbf{A} = (\mathbf{D} + \mathbf{S})\boldsymbol{\Sigma}^{-1}$ is a normal matrix (i.e., if $\mathbf{A}\mathbf{A}^\top = \mathbf{A}^\top\mathbf{A}$) [20]. Then, there exists a unitary matrix $\mathbf{U}$ such that $\mathbf{U}\mathbf{U}^\dagger = \mathbf{I}_{n_p}$ and

$$\mathbf{U}\mathbf{A}\mathbf{U}^\dagger = \mathbf{U}\mathbf{A}^\top\mathbf{U}^\dagger = \mathrm{diag}(\lambda_1, \lambda_2, \ldots, \lambda_{n_p}), \tag{D.8}$$

which, for all $t \geq t'$, yields

$$\mathbf{C}(t, t') = \mathbf{U}^\dagger\mathbf{G}(t, t')\mathbf{U} \tag{D.9}$$

for

$$[\mathbf{G}(t, t')]_{ij} = 2\frac{e^{-\lambda_i|t-t'|} - e^{-\lambda_i t - \lambda_j t'}}{\lambda_i + \lambda_j}[\mathbf{B}\mathbf{B}^\top]_{ij}. \tag{D.10}$$

We will measure the rate of convergence of the ensemble distribution to the stationary distribution in the Kullback-Leibler (KL) divergence and 2-Wasserstein distance for several choices of $\mathbf{D}$. We note that this is not the same measure as considered in our numerical simulations, where we examine the distribution of samples over time within a single trajectory, and measure the mean 2-Wasserstein distance between univariate marginals. As noted in Appendix E, an organism must usually make estimates based on the distribution of samples over a single trajectory. However, even at equilibrium, it is challenging to analytically characterize the 2-Wasserstein distance between samples from a Gaussian distribution and the underlying population distribution [23]. Therefore, we will consider the ensemble distribution at a given time. We denote the KL divergence and 2-Wasserstein distances as a function of time by $K(t)$ and $W_2(t)$, respectively. As all distributions of interest are Gaussian, we have the relatively simple formulas

$$K(t) = \frac{1}{2}\left[\mathrm{tr}\,\boldsymbol{\Sigma}^{-1}\mathbf{C}(t) - n_p + \log\frac{\det\boldsymbol{\Sigma}}{\det\mathbf{C}(t)}\right] \tag{D.11}$$

and

$$W_2(t)^2 = \mathrm{tr}\left[\mathbf{C}(t) + \boldsymbol{\Sigma} - 2(\boldsymbol{\Sigma}^{1/2}\mathbf{C}(t)\boldsymbol{\Sigma}^{1/2})^{1/2}\right]. \tag{D.12}$$

### D.1 Naïve Langevin dynamics

We first consider naïve Langevin sampling with $\mathbf{D} = \mathbf{I}_{n_p}$ and $\mathbf{S} = \mathbf{0}$. In this case, $\mathbf{A} = (\mathbf{D} + \mathbf{S})\boldsymbol{\Sigma}^{-1} = \boldsymbol{\Sigma}^{-1}$ is symmetric and therefore normal, hence, letting the diagonalization of $\boldsymbol{\Sigma}$ be

$$\mathbf{U}\boldsymbol{\Sigma}\mathbf{U}^\dagger = \mathrm{diag}(\sigma_1, \ldots, \sigma_{n_n}), \tag{D.13}$$

the result above yields

$$\mathbf{C}(t, t') = \mathbf{U}^\dagger\mathbf{G}(t, t')\mathbf{U} \tag{D.14}$$

for

$$[\mathbf{G}(t, t')]_{ij} = \sigma_i(e^{-|t-t'|/\sigma_i} - e^{-(t+t')/\sigma_i})\delta_{ij} \tag{D.15}$$

for all $t \geq t'$. In particular, equal-time covariances are governed by

$$[\mathbf{G}(t, t)]_{ij} = \sigma_i(1 - e^{-2t/\sigma_i})\delta_{ij}. \tag{D.16}$$

In matrix form,

$$\mathbf{C}(t, t') = \boldsymbol{\Sigma}(e^{-\boldsymbol{\Sigma}^{-1}|t-t'|} - e^{-\boldsymbol{\Sigma}^{-1}(t+t')}) \tag{D.17}$$

and

$$\mathbf{C}(t) = (\mathbf{I}_{n_p} - e^{-2\boldsymbol{\Sigma}^{-1}t})\boldsymbol{\Sigma}. \tag{D.18}$$

This choice yields stationary covariance

$$\mathbf{C}_s(t - t') = e^{-\boldsymbol{\Sigma}^{-1}|t-t'|}\boldsymbol{\Sigma} \tag{D.19}$$

hence large eigenvalues of $\boldsymbol{\Sigma}$ will introduce long autocorrelation timescales.

In this case, $K(t)$ and $W_2(t)$ are simple to compute thanks to the fact that $\boldsymbol{\Sigma}$ commutes with $\mathbf{C}(t)$, yielding

$$K(t) = -\frac{1}{2}\left[\operatorname{tr} e^{-2\boldsymbol{\Sigma}^{-1}t} + \log\det(\mathbf{I}_{n_p} - e^{-2\boldsymbol{\Sigma}^{-1}t})\right]. \tag{D.20}$$

and

$$W_2(t) = \sqrt{\operatorname{tr}\left\{\boldsymbol{\Sigma}\left[\mathbf{I}_{n_p} - (\mathbf{I}_{n_p} - e^{-2\boldsymbol{\Sigma}^{-1}t})^{1/2}\right]^2\right\}}. \tag{D.21}$$

In terms of the eigenvalues $\sigma_1, \ldots, \sigma_n$ of $\boldsymbol{\Sigma}$, these distances are

$$K(t) = -\frac{1}{2}\sum_{i=1}^{n_p}\left[e^{-2t/\sigma_i} + \log\det(1 - e^{-2t/\sigma_i})\right]. \tag{D.22}$$

and

$$W_2(t) = \sqrt{\sum_{i=1}^{n_p}\sigma_i\left[1 - (1 - e^{-2t/\sigma_i})^{1/2}\right]^2}. \tag{D.23}$$

## D.2 Sampling in the space of natural parameters

We now consider sampling in the space of natural parameters, with $\mathbf{D} = \boldsymbol{\Sigma}$ and $\mathbf{S} = \mathbf{0}$. In this case, $\mathbf{A} = \mathbf{D}\boldsymbol{\Sigma}^{-1} = \mathbf{I}_{n_p}$ is trivially normal, hence, for $t \geq t'$,

$$\mathbf{C}(t, t') = (e^{-|t-t'|} - e^{-(t+t')})\boldsymbol{\Sigma}, \tag{D.24}$$

with

$$\mathbf{C}(t) = (1 - e^{-2t})\boldsymbol{\Sigma} \tag{D.25}$$

in particular. In this case, the stationary covariance is

$$\mathbf{C}_s(t - t') = e^{-|t-t'|}\boldsymbol{\Sigma}, \tag{D.26}$$

hence the stationary autocorrelation timescale will be independent of the spectrum of $\boldsymbol{\Sigma}$.

This setup can also easily be generalized to the case in which $\mathbf{S} \neq \mathbf{0}$. In this case, $\mathbf{A} = (\mathbf{D}+\mathbf{S})\boldsymbol{\Sigma}^{-1} = \mathbf{I}_{n_p} + \mathbf{S}\boldsymbol{\Sigma}^{-1}$ is not in general a normal matrix. However, we can evaluate the covariance function by exploiting the particular structure of the problem. Factoring out the terms involving the identity matrix and using the skew-symmetry of $\mathbf{S}$, we have

$$\mathbf{C}(t, t') = 2\int_0^{\min(t,t')} ds\, e^{-\mathbf{S}\boldsymbol{\Sigma}^{-1}(t-s)}\boldsymbol{\Sigma} e^{\boldsymbol{\Sigma}^{-1}\mathbf{S}(t'-s)}e^{-(t-s)-(t'-s)}. \tag{D.27}$$

We now observe that

$$\boldsymbol{\Sigma}e^{\boldsymbol{\Sigma}^{-1}\mathbf{S}(t'-s)} = \sum_{j=0}^{\infty}\frac{(t'-s)^j}{j!}\boldsymbol{\Sigma}(\boldsymbol{\Sigma}^{-1}\mathbf{S})^j \tag{D.28}$$

$$= \sum_{j=0}^{\infty}\frac{(t'-s)^j}{j!}(\mathbf{S}\boldsymbol{\Sigma}^{-1})^j\boldsymbol{\Sigma} \tag{D.29}$$

$$= e^{\mathbf{S}\boldsymbol{\Sigma}^{-1}(t'-s)}\boldsymbol{\Sigma}, \tag{D.30}$$

hence $e^{-\mathbf{S}\boldsymbol{\Sigma}^{-1}(t-s)}\boldsymbol{\Sigma}e^{\mathbf{S}\boldsymbol{\Sigma}^{-1}(t'-s)} = e^{-\mathbf{S}\boldsymbol{\Sigma}^{-1}(t-s)}e^{\mathbf{S}\boldsymbol{\Sigma}^{-1}(t'-s)}\boldsymbol{\Sigma} = e^{-\mathbf{S}\boldsymbol{\Sigma}^{-1}(t-t')}\boldsymbol{\Sigma}$, and therefore

$$\mathbf{C}(t, t') = 2\int_0^{\min(t,t')} ds\, e^{-(t-s)-(t'-s)}e^{-\mathbf{S}\boldsymbol{\Sigma}^{-1}(t-t')}\boldsymbol{\Sigma}. \tag{D.31}$$

The equal-time covariance is then

$$\mathbf{C}(t) = 2\int_0^t ds\, e^{-2(t-s)}\boldsymbol{\Sigma} \tag{D.32}$$

$$= (1 - e^{-2t})\boldsymbol{\Sigma}, \tag{D.33}$$

as in the case $\mathbf{S} = \mathbf{0}$. Thus, the rate of convergence of the ensemble sampling distribution to the stationary distribution will be unaffected if we add this skew-symmetric term. With the addition of the skew-symmetric term, the stationary covariance is

$$\mathbf{C}_s(t - t') = e^{-|t-t'|}e^{-\mathbf{S}\mathbf{\Sigma}^{-1}(t-t')}\mathbf{\Sigma}, \tag{D.34}$$

which reflects the fact that this term introduces non-reversible dynamics. The spectrum of this matrix is, however, not easy to analyze in general.

Once again, $K(t)$ and $W_2(t)$ are easy to compute thanks to the fact that $\mathbf{C}(t)$ commutes with $\mathbf{\Sigma}$, yielding

$$K(t) = -\frac{n}{2}\left[e^{-2t} + \log(1 - e^{-2t})\right]. \tag{D.35}$$

and

$$W_2(t) = \sqrt{\operatorname{tr}\mathbf{\Sigma}}\left[1 - (1 - e^{-2t})^{1/2}\right]. \tag{D.36}$$

Comparing the $W_2$ distance for naïve Langevin sampling (D.23) with (D.36), we can see that sampling in the space of natural parameters eliminates the sensitivity of convergence speed to large eigenvalues of the covariance matrix. Moreover, comparing the stationary cross-covariance at timelag $\tau \equiv t - t'$ for naïve Langevin sampling, $\mathbf{C}_s(\tau) = e^{-\mathbf{\Sigma}^{-1}|\tau|}\mathbf{\Sigma}$, to that for sampling in the natural space, $\mathbf{C}_s(\tau) = e^{-|\tau|}\mathbf{\Sigma}$, we can see that the same qualitative difference is present. We note that Hennequin et al. [10] focused on the speed of decay in $\mathbf{C}_s(\tau)$. Therefore, in this simple setting, there is a clear intuitive picture of why natural gradients enable fast sampling.

# E  Numerical methods and supplementary figures

In this appendix, we describe our numerical methods and include supplementary figures. All simulations were run in MATLAB 9.10 (R2021a) or 9.12 (R2022a) (The MathWorks, Natick, MA, USA) on desktop workstations (CPU: Intel i9-9900K or Xeon W-2145, 64GB RAM). They were not computationally intensive, and required less than 24 hours of compute time in total. The code used to generate all figures is available from GitHub: https://github.com/Pehlevan-Group/FastSpikingSampler.

For the sweeps in $\rho$ we tested 100 values of $\rho \in [0; 0.99]$. For the dimension sweeps we varied $n_p \in [2, 4, 8, 16, 32, 64]$ and $k = \frac{n_n}{n_p} \in [1, 5, 10, 20]$ is the number of neurons per parameters. Here, we showed results for $k = 10$. As expected, sampling failed in the case of $k = 1$ as the sign constraint introduced by spiking restricts the network to efficiently sample only one half on the values for each parameter but results were qualitatively similar for $k = n_n/n_p \in [5, 10, 20]$.

In Figures 2, E.2, E.3, 4, and E.4, convergence statistics are computed based on distributions of samples over time, that is, the values visited by the sampler over the course of a single trial. Note that this is different than many machine learning studies, which instead consider distributions across realizations at a single timepoint. However, for an organism, probabilistic inference must be performed within a single trial. Because estimation of the full 2-Wasserstein distance between high-dimensional distributions is computationally expensive [24], we instead computed the mean across dimensions of the 2-Wasserstein distances between the marginals of the sampling and target distributions.

For Figures 2, E.2 and E.3, we sampled with a discretization timestep of $\Delta = 10^{-4}$ s and a membrane time constant of $\tau_m = 20$ ms. $\mathbf{\Gamma}$ by drawing independent and identically distributed Gaussian element. We uses $\tau_s = 0.01\tau_m$ and scaled the elastic net regularization parameters using $\alpha = \lambda = \sqrt{n_n}$.

For Figures 3, 4, and E.4, we simulated a Metropolis-Hastings sampling network with a discretization timestep of $\Delta = 10^{-5}$ s and a membrane time constant of $\tau_m = 20$ ms. In the naïve case, we generated the readout matrices as $\mathbf{\Gamma} = [-\mathbf{M}, \mathbf{M}]$ for random matrices $\mathbf{M}$ with independent and identically distributed Gaussian elements. To perform sampling in approximately the natural space, we chose $\mathbf{\Gamma} = \mathbf{\Sigma}^{1/2}[-\mathbf{M}, \mathbf{M}]$ for $\mathbf{M}$ a random matrix with i.i.d. Gaussian elements. For the dimension sweeps in 4c and E.4b, we scale the variance of the elements of $\mathbf{M}$ to be $1/n_p$. We probe the sensitivity of the sampler to the variance of the random matrix in Figure B.1, showing

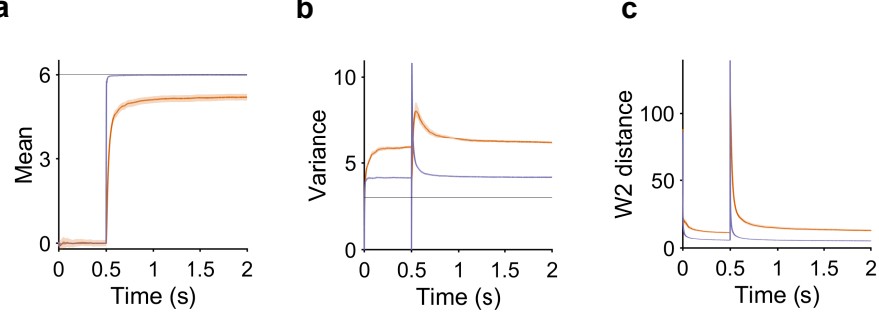

Figure E.1: Time course of the inference statistics in EBNs implementing Langevin sampling using naïve and natural geometry. There is initially no input (target mean $\mu = 0$). At time $t = 0.5s$, the stimulus appears (target mean $\mu = 6$). The network parameters are $\rho = 0.8$, $n_p = 20$ and $n_n = 200$. This corresponds to network parameters for which the difference between the naïve and natural geometry starts to be sizeable (See Figures 2.c and E.3.a). **a.** The inferred mean converges faster for the natural geometry and there is a steady state error in the naïve implementation. **b.** The transient in the value of the estimated variance at stimulus onset ($t = 0.5s$) and the steady state error are larger and the transient relaxes to baseline more slowly in the naïve implementation than with the natural geometry. **c.** The transient in the $W_2$ distance to the target distribution at stimulus onset ($t = 0.5s$) and the steady state $W_2$ distance are larger and the transient relaxes to baseline more slowly in the naïve implementation than with the natural geometry. Shaded error patches show 95% confidence intervals computed via bootstrapping over 100 realizations in all panels

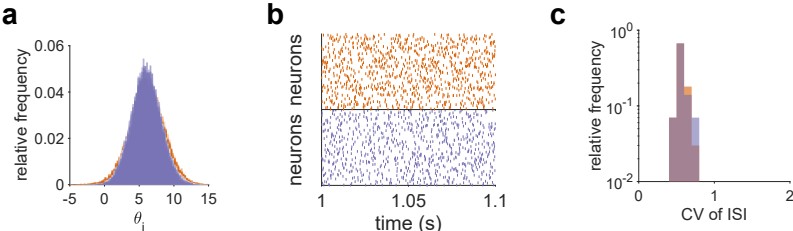

Figure E.2: Additional statistics of spike trains in EBNs implementing Langevin sampling using naïve and natural geometry. **a.** Marginal distribution of $\theta_i$ after stimulus onset. **b.** Example raster plots over a 100 ms window for naïve and natural geometry. **c.** The distribution of coefficients of variation of ISIs across neurons.

that its performance is robust within some range of variances. The distributions in Figure 3b were generated using 1000 realizations of the randomness in the proposal and accept/reject steps for a single realization of the random matrix **M**. Statistics in Figures 4 and E.4 were computed across 100 realizations of the random matrix **M** and of the randomness in the proposal and accept/reject steps.

## Supplemental references

[1] Gareth O Roberts and Richard L Tweedie. Exponential convergence of Langevin distributions and their discrete approximations. *Bernoulli*, pages 341–363, 1996. doi: 10.2307/3318418.

[2] Ruqi Zhang, A Feder Cooper, and Christopher De Sa. AMAGOLD: Amortized Metropolis adjustment for efficient stochastic gradient MCMC. In *International Conference on Artificial Intelligence and Statistics*, pages 2142–2152. PMLR, 2020. URL https://proceedings.mlr.press/v108/zhang20e.html.

[3] Martin Boerlin, Christian K Machens, and Sophie Denève. Predictive coding of dynamical variables in balanced spiking networks. *PLoS Computational Biology*, 9(11):e1003258, 2013. doi: 10.1371/journal.pcbi.1003258.

[4] Roger A Horn and Charles R Johnson. *Matrix Analysis*. Cambridge University Press, 2012. doi: https://doi.org/10.1017/CBO9780511810817.

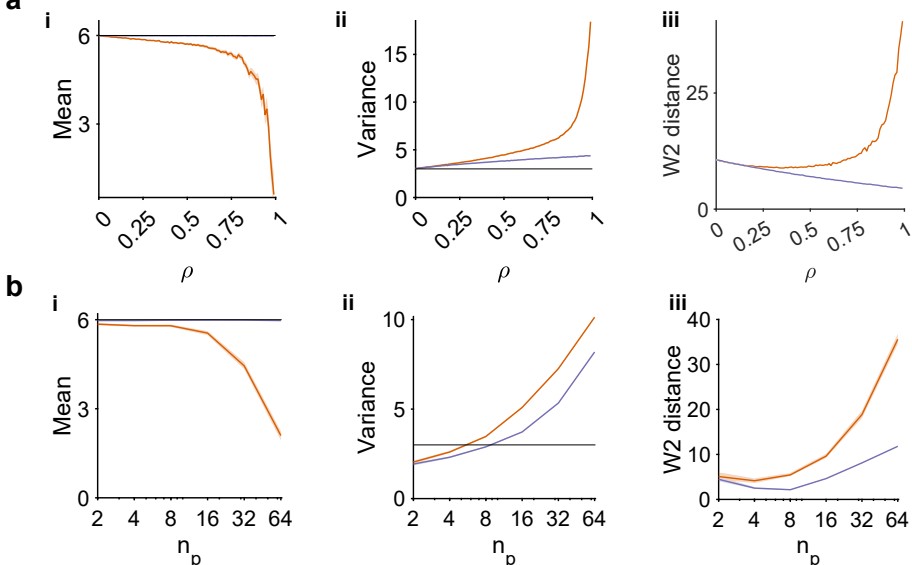

Figure E.3: Inference statistics for sampling in efficient balanced networks at steady state (using full 1.5s after stimulus onset). The setup is as in Figure 2. **a**. Comparison of performance in the 50 ms following stimulus onset (after $t = 0.5$ s) between naïve and natural geometry for varying $\rho$ **a.i**. The estimate of the mean collapses towards zero with increasing $\rho$ for naïve geometry, **a.ii**. The estimated variance increases catastrophically with $\rho$ for naïve geometry, but only mildly for natural geometry. **a.iii**. Inference accuracy as measured by the mean marginal 2-Wasserstein distance across dimensions decreases for naïve. **b**. Similar analysis when varying $n_p$ for the **d.i**. Mean, **d.ii**. Variance and **b.iii**. mean 2-Wasserstein distance. Shaded error patches show 95% confidence intervals computed via bootstrapping over 100 realizations in all panels; see Appendix E for further details. See Appendix E for detailed numerical methods.

[5] Cristina Savin and Sophie Denève. Spatio-temporal representations of uncertainty in spiking neural networks. In Z. Ghahramani, M. Welling, C. Cortes, N. Lawrence, and K.Q. Weinberger, editors, *Advances in Neural Information Processing Systems*, volume 27, pages 2024–2032. Curran Associates, Inc., 2014. URL https://proceedings.neurips.cc/paper/2014/hash/4e2545f819e67f0615003dd7e04a6087-Abstract.html.

[6] Sophie Denève and Christian K Machens. Efficient codes and balanced networks. *Nature Neuroscience*, 19(3):375–382, 2016. doi: 10.1038/nn.4243.

[7] Sophie Denève, Alireza Alemi, and Ralph Bourdoukan. The brain as an efficient and robust adaptive learner. *Neuron*, 94(5):969–977, 2017. doi: 10.1016/j.neuron.2017.05.016.

[8] Camille E Rullán Buxó and Jonathan W Pillow. Poisson balanced spiking networks. *PLoS Computational Biology*, 16(11):e1008261, 2020. doi: 10.1371/journal.pcbi.1008261.

[9] Nuno Calaim, Florian A Dehmelt, Pedro J Gonçalves, and Christian K Machens. The geometry of robustness in spiking neural networks. *eLife*, 11:e73276, may 2022. ISSN 2050-084X. doi: 10.7554/eLife.73276. URL https://doi.org/10.7554/eLife.73276.

[10] Guillaume Hennequin, Laurence Aitchison, and Mate Lengyel. Fast sampling-based inference in balanced neuronal networks. In Z. Ghahramani, M. Welling, C. Cortes, N. Lawrence, and K.Q. Weinberger, editors, *Advances in Neural Information Processing Systems*, volume 27, pages 2240–2248. Curran Associates, Inc., 2014. URL https://papers.nips.cc/paper/2014/hash/a7d8ae4569120b5bec12e7b6e9648b86-Abstract.html.

[11] Yi-An Ma, Tianqi Chen, and Emily Fox. A complete recipe for stochastic gradient MCMC. In C. Cortes, N. Lawrence, D. Lee, M. Sugiyama, and R. Garnett, editors, *Advances in Neural Information Processing Systems*, volume 28. Curran Associates, Inc., 2015. URL https://papers.nips.cc/paper/2015/hash/9a4400501febb2a95e79248486a5f6d3-Abstract.html.

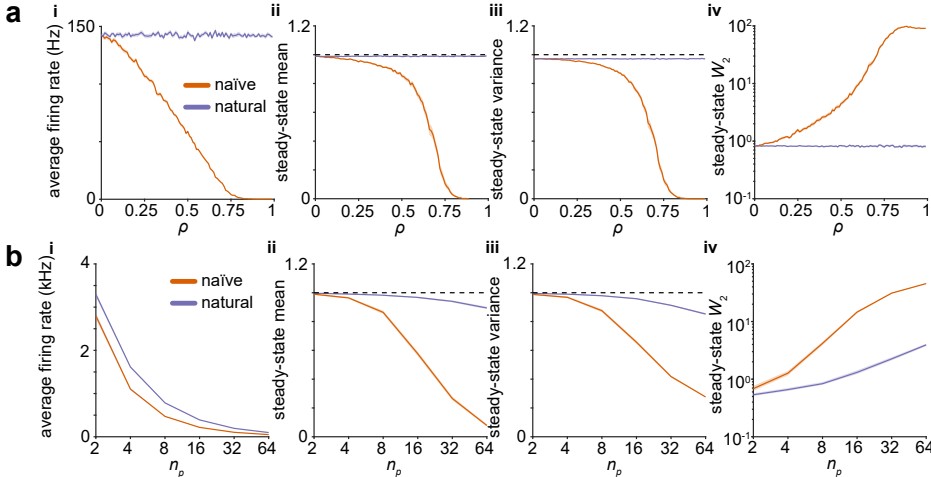

Figure E.4: Steady-state statistics for the Metropolis-Hastings sampler of Figure 3. **a**. Sampling from strongly-equicorrelated Gaussians in $n_p = 10$ dimensions using a network of $n_n = 100$ neurons requires careful choice of geometry. The stimulus setup is as in Figure 3. **a.i**. At strong correlations $\rho$, spiking is suppressed in networks with naïve geometry, but not when the natural geometry is used. This panel is identical to Figure 3b.i. **a.ii**. Dimension-averaged estimate of the mean signal over the entire stimulus interval. **a.iii**. As in **ii**, but for the variance. **a.iv**. As in **ii**, but for the 2-Wasserstein distance. **b**. Sampling in high dimensions requires careful choice of geometry. **b.i**. At moderately strong correlations $\rho = 0.75$ and high dimensions, spiking is suppressed in networks with naïve geometry, but not when the natural geometry is used. Here, we use 10 neurons per parameter, i.e., $n_n = 10n_p$. **b.ii**. Dimension-averaged estimate of the mean signal in the 50 milliseconds after stimulus onset. This panel is identical to Figure 3c.i. **b.iii**. As in **ii**, but for the variance. **b.iv**. As in **ii**, but for the 2-Wasserstein distance. Shaded error patches show 95% confidence intervals computed via bootstrapping over 100 realizations in all panels; see Appendix E for further details.

[12] Shun-Ichi Amari. Natural gradient works efficiently in learning. *Neural Computation*, 10(2): 251–276, 1998. doi: 10.1162/089976698300017746.

[13] Shun-Ichi Amari and Scott C Douglas. Why natural gradient? In *Proceedings of the 1998 IEEE International Conference on Acoustics, Speech and Signal Processing, ICASSP'98 (Cat. No. 98CH36181)*, volume 2, pages 1213–1216. IEEE, 1998. doi: 10.1109/ICASSP.1998.675489.

[14] Mark Girolami and Ben Calderhead. Riemann manifold Langevin and Hamiltonian Monte Carlo methods. *Journal of the Royal Statistical Society: Series B (Statistical Methodology)*, 73 (2):123–214, 2011. doi: 10.1111/j.1467-9868.2010.00765.x.

[15] James Martens and Roger Grosse. Optimizing neural networks with Kronecker-factored approximate curvature. In *International Conference on Machine Learning*, pages 2408–2417. PMLR, 2015. doi: 10.48550/arXiv.1503.05671.

[16] James Martens. New insights and perspectives on the natural gradient method. *Journal of Machine Learning Research*, 21:1–76, 2020. URL https://jmlr.org/papers/v21/17-678.html.

[17] Chii-Ruey Hwang, Shu-Yin Hwang-Ma, and Shuenn-Jyi Sheu. Accelerating diffusions. *The Annals of Applied Probability*, 15(2):1433–1444, 2005. doi: 10.1214/105051605000000025.

[18] Christian P Robert, Víctor Elvira, Nick Tawn, and Changye Wu. Accelerating MCMC algorithms. *Wiley Interdisciplinary Reviews: Computational Statistics*, 10(5):e1435, 2018. doi: 10.1002/wics.1435.

[19] Wei Ji Ma, Jeffrey M Beck, Peter E Latham, and Alexandre Pouget. Bayesian inference with probabilistic population codes. *Nature Neuroscience*, 9(11):1432–1438, 2006. doi: 10.1038/nn1790.

[20] Crispin W Gardiner. *Handbook of stochastic methods*, volume 3. Springer Berlin, 1985.

[21] Bernt Øksendal. *Stochastic differential equations*. Springer, 2003. doi: 10.1007/978-3-642-14394-6.

[22] Claude Godrèche and Jean-Marc Luck. Characterising the nonequilibrium stationary states of Ornstein–Uhlenbeck processes. *Journal of Physics A: Mathematical and Theoretical*, 52(3): 035002, 2018. doi: 10.1088/1751-8121/aaf190.

[23] Thomas Rippl, Axel Munk, and Anja Sturm. Limit laws of the empirical Wasserstein distance: Gaussian distributions. *Journal of Multivariate Analysis*, 151:90–109, 2016. ISSN 0047-259X. doi: https://doi.org/10.1016/j.jmva.2016.06.005. URL https://www.sciencedirect.com/science/article/pii/S0047259X16300446.

[24] Lénaïc Chizat, Pierre Roussillon, Flavien Léger, François-Xavier Vialard, and Gabriel Peyré. Faster Wasserstein distance estimation with the Sinkhorn divergence. In H. Larochelle, M. Ranzato, R. Hadsell, M.F. Balcan, and H. Lin, editors, *Advances in Neural Information Processing Systems*, volume 33, pages 2257–2269. Curran Associates, Inc., 2020. URL https://proceedings.neurips.cc/paper/2020/hash/17f98ddf040204eda0af36a108cbdea4-Abstract.html.