# OpenReview forum: "Natural gradient enables fast sampling in spiking neural networks"
_NeurIPS.cc/2022/Conference — NeurIPS 2022 Accept_

### Official Review · Reviewer_96zE · 2022-07-04

**Rating:** 5
**Confidence:** 4
**Soundness:** 2 fair
**Presentation:** 3 good
**Contribution:** 2 fair

**Summary:**

In this work, the authors demonstrate the possibility of implementing sampling algorithms through spiking neural networks. They provide details on how to realize the metropolis-hastings sampler using the probabilistic spike rules. Overall the work is interesting and extends the potential application of spiking neural networks.

**Questions:**

1. What's the connection between the brain's capacity for probabilistic computation and the ability of SNN for sampling? What's the evidence for the brain to realize its probabilistic computation through the same sampling approach as we do in the Bayesian inference?

2. The implementation needs to accept the spikes probabilistically. How to realize this mechanism in a spiking neural network?

3. Why is it called fast sampling? How is it faster when compared to other methods?

4. What's the infrastructure for the SNN used for sampling?

5. If the probabilistic spiking mechanism is allowed, why do we need such a spiking neural network rather than using a regular artificial neural network for the computation in the sampling procedure?

**Limitations:**

NA.

**Strengths And Weaknesses:**

Strengths:
1. The application is relatively new for SNN.
2. The writing is clear.
Weakness:
1. The motivation for applying SNN for sampling is not convincing enough.
2. The comparison to traditional sampling is not enough.

---

> ### Author Response · Authors · 2022-08-02
> **Response to Reviewer 96zE**
>
> We thank the reviewer and appreciate the reviewer's opinion on our paper. We noted that most of the reviewer's questions are about the high level motivations behind our work, which we clarify below. We therefore ask the reviewer to kindly reconsider their assessment of our work.
>
> *[...]*
>
> #### Questions:
>
> - *What's the connection between the brain's capacity for probabilistic computation and the ability of SNN for sampling?*
>
>     As we describe in Lines 18-23 of the Introduction, there is ample evidence that the brain performs probabilistic computations:
>
>     > Neural circuits perform probabilistic computations at the sensory, motor and cognitive levels (Knill and Pouget 2004, Körding and Wolpert 2004, Fiser et al. 2010, Ott, Masset, and Kepecs 2018). From abstract representations of decision confidence (Masset et al. 2020) to estimates of sensory uncertainty in visual cortex (van Bergen et al. 2015, Festa et al. 2021), evidence of probabilistic representations can be found at all levels of the cortical processing hierarchy (Pouget, Drugowitsch, and Kepecs 2016).
>
>     As sampling is one of the most promising implementation of probabilistic computation at scale in the ML literature, it is natural to explore how biologically neural networks could efficiently implement sampling algorithms.
>
>
> - *What's the evidence for the brain to realize its probabilistic computation through the same sampling approach as we do in the Bayesian inference?*
>
>     Thank you for this question. Recent work by Echeveste et al., (Nature Neuroscience, 2020) has shown that sampling codes predict specific properties of neurons in visual cortex. We previously mentioned this paper briefly in the Introduction; we now describe it in more detail. The relevant sentence reads as follows:
>
>     > Moreover, they predict specific properties of neural responses in visual cortex, including changes in Fano factor and frequency of oscillations with tuning and stimulus intensity (Echeveste et al., 2020).
>
> - *The implementation needs to accept the spikes probabilistically. How to realize this mechanism in a spiking neural network?*
>
>     Thank you for this question. Including this stochasticity actually improves the biological plausibility of our proposed network. In biological neurons, the stochasticity of the spike generation and synaptic release mechanisms are well established phenomena through decades of experimental work. The probabilistic acceptance of spike proposals in our network may be realized in biology by these mechanisms. In the submitted manuscript, we mention this point in Lines 307-315 of the Discussion:
>
>     > In biological spiking networks, probabilistic spike emission and probabilistic synaptic release are natural sources of stochasticity (Del Castillo and Katz, 1954; de Ruyter van Steveninck et al., 1997; Mongillo, Barak, and Tsodkys, 2008; Rusakov, Savtchenko, and Latham, 2020).
>
>     We are happy to extend our discussion of this point if the reviewer believes it is necessary.
>
>
> - *Why is it called fast sampling? How is it faster when compared to other methods?*
>
>     We use the term "fast sampling" as the networks we propose are able to sample high-dimensional posterior distribution after 50ms given biologically plausible constraint on the timescales of single neurons' membrane potentials (see Lines 237-241). To further illustrate how using natural gradients enables faster sampling than geometry-naïve samplers, we added Appendix Figure E.1 (\url{https://figshare.com/s/15ccd407eb0ba614d23d}), in which we show the dynamics of the estimate mean, estimate variance and Wasserstein-2 distance to the target distribution as a function of time. These simulations highlight how leveraging the geometry improves both the speed and the accuracy of the inference.
>
>
> - *What's the infrastructure for the SNN used for sampling?*
>
>     We apologize for our confusion, but unfortunately we could not understand the referee's question. We introduce the SNNs used for sampling in Section 2, with detailed derivations in Appendices B and C. Note that the networks here are simulated rather than implemented in neuromorphic hardware as we are interested in the computational principles of these SNNs. We might be misinterpreting the reviewer's question, please ask if more information is needed.
>
>
> - *If the probabilistic spiking mechanism is allowed, why do we need such a spiking neural network rather than using a regular artificial neural network for the computation in the sampling procedure?*
>
>     The probabilistic spiking mechanism only applies to the generation of the spike. The spikes themselves are still binary events and neurons only communicate with each other through these discrete spikes. Our network therefore strongly differs from regular neural networks and carries all the associated challenges and benefits of SNNs. We might be misinterpreting the reviewer's question; please ask if more information is needed.
>
> *[...]*

---

> > ### Comment · Reviewer_96zE · 2022-08-06
> > **Some clarification on my comments**
> >
> > The author is right that I mainly raised some  high-level motivations and reasonability about this paper as I believe that this is more important for a new question. Based on the reviewer's response, I still have some puzzles.
> >
> > 1. I did not doubt that the brain has the ability of probabilistic computation. My concern here is what's the relationship between the brain's probabilistic computation and the framework of SNN sampling in the current paper? There is a gap between this two part. To me, it is not trivial given the references.
> >
> > 2. In terms of the network infrastructure, do you mean you used  a recurrent setup as shown in Figure 1? If it is true, what is the time length T to achieve sufficient accurate samples? What's the energy cost compared to using traditional sampling approaches? Can the authors provide any quantitative results to support the superiority of using SNN here?

---

> > > ### Author Response · Authors · 2022-08-08
> > > **Reply to clarifications from Reviewer 96zE**
> > >
> > > Please find below our detailed responses to the reviewer's clarification points.
> > >
> > > *I did not doubt that the brain has the ability of probabilistic computation. My concern here is what's the relationship between the brain's probabilistic computation and the framework of SNN sampling in the current paper? There is a gap between this two part. To me, it is not trivial given the references.*
> > >
> > > It is well know that both at steady state and across trials there is variability in the firing of neurons. As we highlight in the introduction, the hypothesis that this variability in neural firing is a signature of sampling  from a posterior in a MCMC framework gained traction with the work of Hoyer & Hyvarinen (NeurIPS, 2002). As we point out in the introduction, the advantage of the sampling hypothesis is that it relies on a body of work in machine learning and statistics that shows they could in principle perform computations for high-dimensional posteriors. In contrast, while other proposed schemes perform well on simple toy models (e.g. cue combination between two sensory input streams), their is little work scaling them to useful models in machine learning applications.  Since then many papers have explored the sampling  hypothesis but few have explored the ability of such schemes to perform probabilistic computations at a speed consistent with perception (~$50ms$) and for high-dimensional posterior distributions. As the brain is a spiking neural network, it is natural to explore potential sampling implementations in SNNs rather than rate networks.
> > >
> > > We would also like to point out further experimental evidence such as the paper by Berkes and colleagues (Science, 2011, <https://www.science.org/doi/full/10.1126/science.1195870>) which shows in recordings from ferret visual cortex that the space explored by spontaneous activity in the absence of sensory stimulus is consistent with exploration of the prior in sampling algorithms.
> > >
> > > We would be happy to extend the paragraph in the introduction to better introduce this work linking probabilistic computations in the brain to sampling in the in the final version of the paper with the additional content page.
> > >
> > >
> > > *If it is true, what is the time length T to achieve sufficient accurate samples?*
> > >
> > > Figures 2 and 4 show that for both our implementations, the inference at $t=50ms$ after stimulus onset is much improved when using the natural geometry in comparison to the naive geometry. In our revisions, we have also added Appendix Figure E.1  (<https://figshare.com/s/15ccd407eb0ba614d23d>), in which we specifically show the dynamics of the estimated mean, variance and Wasserstein-2 distance to the target distribution as a function of time.
> > >
> > > *What's the energy cost compared to using traditional sampling approaches? Can the authors provide any quantitative results to support the superiority of using SNN here?*
> > >
> > > In biological networks, computing with spikes reduces energy expenditure by 10-100 folds compared to sending continuous signals (Laughlin et al, 1998, Nature Neuroscience <https://www.nature.com/articles/nn0598_36>). Indeed, in most biological networks, non-spiking neurons are usually only located in the earliest stages of sensory processing where signal transduction occurs. Spiking artificial neural networks with similar abilities than ANNs implemented on neuromorphic chips is a long term goal of the field as it is recognized they could offer orders of magnitude improvements in energy efficiency (e.g. Roy et al., Nature, 2019 <https://www.nature.com/articles/s41586-019-1677-2> or Markovic et al, Nature reviews Physics, 2020 <https://www.nature.com/articles/s42254-020-0208-2>).
> > >
> > > Our model is a proposed computational framework and is not tied to a specific hardware implementation. The resulting energy expenditure will be dependent on many factors such as whether the network is simulated on a standard CPU/GPU or directly implemented in a neuromorphic chip.
> > >
> > > We would be happy to include an extended discussion of these points in the final version of the paper with the additional content page.

---

### Official Review · Reviewer_jyfu · 2022-07-11

**Rating:** 6
**Confidence:** 4
**Soundness:** 2 fair
**Presentation:** 2 fair
**Contribution:** 2 fair

**Summary:**

The present paper studies sampling in spiking network models and tries to unify the Langevin sampling in the efficient balanced networks and Metropolis-Hastings sampling in networks with probabilistic spike rules. And it also studies how the geometric structure in the neural code speeds up the sampling.

**Questions:**

Please refer to my comments in the strengths and weaknesses.

**Ethics Review Area:**

["I don’t know"]

**Limitations:**

The limitation of the current framework is the analog-digital conversion (DAC) in sampling a continuous distribution in discrete spiking networks, as mentioned in line 238. This makes people feel that the spiking network is not designed by implementing sampling although the error of DAC can be reduced by considering population geometry or in large number limit. Similar feelings were mentioned by the authors in the paper. I hope to see more discussion of alternative solutions to the DAC error. One possible solution is utilizing the math techniques in probabilistic population code to sample continuous distribution in discrete spikes, where the representation of a continuous distribution in spiking networks comes from the continuously smooth tuning curves of neurons.

**Strengths And Weaknesses:**

The strength of this paper is its unifying nature. And its proposal of a probabilistic spike rule implementing Metropolis-Hastings sampling is novel. The present study did some theoretical derivations in linking the sampling dynamics with neural dynamics, but some of the presentations is not very clear.

### Major

I have a couple of major concerns about this present paper.
-  A strong requirement of neural sampling is that the neural circuit dynamics with fixed parameters is able to sample posteriors with different uncertainties. It is not clear whether this can be achieved in the current framework. The proposed network model seems not able to achieve this if I understood correctly, in that the network parameters in Eq. 12 depend on the mean and the variance (explained by the text in line 160). This implies that to sample distribution with different parameters we need to adjust the network connections.


- As mentioned in Eq. 14 that the D matrix only modifies the temporal dynamics but leaves the sampling distribution unchanged. That means sampling in the Euclidean space and the natural space (including the inverse Fisher information matrix, line 195) should not change the sampling distribution. However, from Fig. 2 and Fig. 3 the two sampling trajectories (blue vs. red) differ a lot and have different variances. I am wondering whether the sampling is correct or not.

### Writing
The writing in sec 2.1 is not clear and I feel difficulty sometimes in following the flow. A lot of questions about the presentation remain.

- It is not clear why the problem from discrete and sign-constrained spikes can be solved by imposing a fine-tuned balancing condition on the readout weights (lines 93-98). More explanations are needed. Also, it is not clear the motivation for choosing a neuron uniformly at random (line 104).

- In Eq. 1, should I interpret the estimate $\hat{\theta}_t$ as a sample at time t? Is the uncertainty of the sampling distribution represented by the fluctuation of $\hat{\theta}_t$ over time? It is better to explicitly mention this in the beginning.

- Eqs. 3 and 4: the argument in the distribution in the denominator of Eq. 4, i.e., $(1-\eta)\Gamma r_{t-1}$ is not consistent with $\hat{\theta}_{t-1}$ in the denominator of Eq. 3.

- It is unclear how Eq. 6 and 10 are related to the undefined generative model which was mentioned in line 71. Is there an implicit assumption of the linear Gaussian generative model because the terms in Eq. 10 are the derivative of quadratic terms. Moreover, it is not clear how \psi is defined at this moment until later I read the text in line 160. Also, in Eq. 6 the input to the neural dynamics is not the observation x defined in the generative model (line 71), but the latent variable $\theta$ directly. More explanations are needed here.

- Line 195: it is not clear what the matrix G is until I read line 232. Please define symbols when they first appear.

---

> ### Author Response · Authors · 2022-08-02
> **Response to Reviewer jyfu (Part 1 of 4)**
>
> *[...]*
>
> #### Major
>
> *[...]*
>
> - *A strong requirement of neural sampling is that the neural circuit dynamics with fixed parameters is able to sample posteriors with different uncertainties. It is not clear whether this can be achieved in the current framework. The proposed network model seems not able to achieve this if I understood correctly, in that the network parameters in Eq. 12 depend on the mean and the variance (explained by the text in line 160). This implies that to sample distribution with different parameters we need to adjust the network connections.*
>
>     Thank you for this comment. Please see our response to Reviewer TEk8's question regarding input dependence, which we copy here:
>     > In our construction, the input to the network appears through the time-varying mean ${\theta}(t)$. In the linear Gaussian model we discuss in Lines 71-80 of Section 2 and detail in Appendix C.2-3, the network infers the expected value of the parameters/latent variables given a sensory input $\textbf{x}$, of which the target mean ${\theta}$ is a linear function (see also Hennequin et al. and Savin and Denève, who use this model). In this simple setting, the covariance structure does not change with the input. The new supplementary Figure E.1, anonymously archived at \url{https://figshare.com/s/15ccd407eb0ba614d23d}, highlights this point. The estimate of the variance is perturbed by stimulus onset (Figure E.1b) but relaxes to its steady state value. Note that this relaxation is both faster and more accurate in the natural geometry condition. For non-Gaussian posteriors, the variance would be changing with the input as well.
>     > Note that the relative timescale of changes in the posterior is the important point. Our result show that given a change in input, the network implementing the inference in the natural space of parameters samples the new posterior within 50 ms for a large range of parameters. As long as the changes in the posterior occur at this speed (the speed of perception), it will be adequately sampled.
>     > If the reviewer deems it necessary, we are happy to include an extended discussion of these points in the Introduction and discussion of the final version of the manuscript; we have not done so thus far because of space constraints.
>
>
> - *As mentioned in Eq. 14 that the D matrix only modifies the temporal dynamics but leaves the sampling distribution unchanged. That means sampling in the Euclidean space and the natural space (including the inverse Fisher information matrix, line 195) should not change the sampling distribution. However, from Fig. 2 and Fig. 3 the two sampling trajectories (blue vs. red) differ a lot and have different variances. I am wondering whether the sampling is correct or not.*
>
>     Thank you for this question. The discrepancies you observe are expected if one considers the following factors. As noted in Lines 240-244, the overestimation of variance observed in Figure 2 is expected due to discretization error even for non-spiking Langevin samplers without accept-reject steps; the further discretization due to spiking exacerbates this effect (Neal, 1993; Savin and Denéve 2014). We find empirically that this overestimation is less for natural geometry. In regards to Figures 3 and 4, we show analytically in Appendix B.5 that one does not expect networks with naïvely chosen weights to correctly sample strongly correlated Gaussian distributions (see Lines 267-271). In brief, the firing rate should tend to zero as the correlation goes to one, as all proposed spikes will be rejected. Therefore, in these figures, the naïve spiking samplers are failing to sample from the correct distribution. Using natural geometry eliminates this problem. With these considerations in mind, the noted discrepancies demonstrate the advantages of using natural gradients.
>
>     We remark also that the work of Ma et al (2015) shows only that different choices of $\mathbf{D}$ yield the same stationary distribution. Therefore, one does not expect the sampling distribution at a short, finite time (as studied in Figures 2, 3, and 4) to be unchanged by modifying $\mathbf{D}$ (see also Figure E.1). Indeed, accelerating the rate of convergence to the stationary distribution is precisely the goal of the complete recipe; our new analysis of linear rate networks illustrates this point. This equivalence of stationary distributions holds for the continuous-time dynamics, and is not guaranteed to hold after discretization.

---

> > ### Author Response · Authors · 2022-08-02
> > **Response to Reviewer jyfu (Part 2 of 4)**
> >
> > #### Writing
> >
> > *[...]*
> >
> >
> > - *It is not clear why the problem from discrete and sign-constrained spikes can be solved by imposing a fine-tuned balancing condition on the readout weights (lines 93-98). More explanations are needed. Also, it is not clear the motivation for choosing a neuron uniformly at random (line 104).*
> >
> >     Thank you for this comment. The balancing condition makes the proposal distribution symmetric, but does not resolve the issue of discretization. We have added the following sentence to Lines 99-101 of the updated manuscript to clarify this point:
> >
> >     > To obtain a symmetric proposal distribution with sign-constrained spikes, we assume that the network is divided into two equally-sized populations with equal and opposite readout weights.
> >
> >     Please see also our response to Reviewer Pjbs' question on the asynchronous update rule for further discussion of this point, which we quote below::
> >
> >     > Thank you for these comments. With regards to the membrane time constant, we note that we are assuming that the decay constant in discrete time $\eta$ is small. In terms of the membrane time constant $\tau_{m}$ and timestep between spike proposals $\Delta$, we have $\eta = \Delta/\tau_{m}$, hence what we assume is that the membrane time constant is long relative to the interval between updates, not that it is long in an absolute sense. Indeed, in Section 2.2, we take the continuum limit $\Delta \downarrow 0$ in which spike proposals are made infinitely often. Therefore, our model does not require membrane time constants $\tau_{m}$ that are longer than is plausible biologically.
> >     The fact that only one neuron is allowed to spike at a time is not a severe constraint relative to prior works, as it is standard in studies of efficient balanced networks (EBNs). As mentioned in Lines 280-288 of the Discussion, the constraint that only one neuron is allowed to spike at each timestep is present in previous work on efficient balanced networks (Boerlin et al 2013, Savin and Denève 2014), and some mechanism for constraining firing rate is present in all works on the subject (Rullán Buxó and Pillow 2021, Calaim et al 2022). As noted there, relaxing these constraints will be an important objective for future work.
> >
> >
> > - *In Eq. 1, should I interpret the estimate $\hat{\theta}\_{t}$ as a sample at time t? Is the uncertainty of the sampling distribution represented by the fluctuation of $\hat{\theta}\_{t}$ over time? It is better to explicitly mention this in the beginning.*
> >
> >     We are sorry for the confusing notation. You are correct in interpreting $\hat{\theta}\_{t}$ as a sample at time $t$. To make this more transparent (and avoid possible confusion between samples from the distribution and an estimate of its mean), we now denote samples by $z\_{t}$ rather than $\hat{\theta}\_{t}$.
> >
> >
> > - *Eqs. 3 and 4: the argument in the distribution in the denominator of Eq. 4, i.e., $(1-\eta) \Gamma r\_{t-1}$ is not consistent with $\hat{\theta}\_{t-1}$ in the denominator of Eq. 3.*
> >
> >     Thank you for this comment. You are correct in noting that the argument of the denominator of the acceptance ratio of the model with decay ($(1-\eta) \mathbf{z}\_{t-1}$) does not match that for the perfect integrator model ($\mathbf{z}\_{t}$). However, as noted in the main text (Lines 115-118 of the originally submitted manuscript) and in Appendix B, this is an intentional design choice, not a mistake. We have updated the discussion around these equations to further clarify this point, which now reads as follows:
> >
> >     > [B]y comparing the likelihood of the proposal, $P[(1-\eta) \mathbf{z}\_{t-1} + {\Gamma} \mathbf{e}\_{j}]$, to the likelihood of the next state without the proposed spike but with the decay $P[(1-\eta) \mathbf{z}\_{t-1}]$ (instead of the likelihood of the current state $P[\mathbf{z}\_{t-1}]$ as in the Metropolis-Hastings algorithm), this choice implements a sort of look-ahead step that should allow the algorithm to partially compensate for the decay in the rate.
> >
> >     We hope that this clarification helps address your concern.

---

> > > ### Author Response · Authors · 2022-08-02
> > > **Response to Reviewer jyfu (Part 3 of 4)**
> > >
> > > #### Writing (Part 2 of 2)
> > >
> > > - *It is unclear how Eq. 6 and 10 are related to the undefined generative model which was mentioned in line 71. Is there an implicit assumption of the linear Gaussian generative model because the terms in Eq. 10 are the derivative of quadratic terms. Moreover, it is not clear how $\psi$ is defined at this moment until later I read the text in line 160. Also, in Eq. 6 the input to the neural dynamics is not the observation x defined in the generative model (line 71), but the latent variable  directly. More explanations are needed here.*
> > >
> > >     Thank you for these suggestions, and we apologize for the confusion. As noted in Lines 77-78 (where $\Psi$ was first defined) of the originally submitted manuscript, and again in Line 160, the goal of Section 2.1 is to build a network that (approximately) samples from a general Gaussian distribution with mean ${\theta}$ and covariance ${\Psi}$. It is to this distribution that the equations you mention refer; we now state this point more explicitly in Lines 77-78. We included the discussion on lines 68-75 in the hope of providing a more concrete point of reference, and regret that its link to the abstract discussion in the remainder of Section 2 was unclear. We have revised this prose as follows:
> > >
> > >     > In this work, we will keep our discussion quite general, and state our results for sampling from a generic Gaussian distribution. However, the problem we aim to solve can be given a concrete interpretation in a neuroscience context, and could also be extended to non-Gaussian distributions. The goal of a neural network performing probabilistic inference is to estimate a posterior distribution $P({\theta} | \mathbf{x})$ over $n_p$ latent variables (or parameters) ${\theta}$ given an input $\mathbf{x}$ (Figure 1). The input could correspond to the activity of sensory neurons in early sensory processing (e.g. input onto ganglion cells in the retina or onto mitral cells in the olfactory bulb) or inputs into a cortical column that linearly sense features in the environment through an affinity matrix. We provide a detailed discussion of this linear Gaussian model in Appendix C. In the rest of the paper, we will usually abbreviate the distribution from which we want to sample as $P({\theta})$, rather than writing $P({\theta} | \mathbf{x})$.
> > >
> > > - *Line 195: it is not clear what the matrix G is until I read line 232. Please define symbols when they first appear.*
> > >
> > >     Thank you for this comment, and we apologize for the confusion. The sentence to which you refer should read "Samplers based on Riemannian geometry can be designed by choosing $\mathbf{D}$ to be the inverse of the Fisher information matrix $\mathbf{G}$ (or an approximation thereof), yielding a preconditioned gradient ${\nabla}_{\textrm{nat}} U = \mathbf{G}^{-1} {\nabla} U$;" the initial usage of $\mathbf{G}$ was ommitted due to a typo in our submitted manuscript.
> > >
> > > *[...]*

---

> > > > ### Author Response · Authors · 2022-08-02
> > > > **Response to Reviewer jyfu (Part 4 of 4)**
> > > >
> > > > #### Limitations
> > > >
> > > > *The limitation of the current framework is the analog-digital conversion (DAC) in sampling a continuous distribution in discrete spiking networks, as mentioned in line 238. This makes people feel that the spiking network is not designed by implementing sampling although the error of DAC can be reduced by considering population geometry or in large number limit. Similar feelings were mentioned by the authors in the paper. I hope to see more discussion of alternative solutions to the DAC error. One possible solution is utilizing the math techniques in probabilistic population code to sample continuous distribution in discrete spikes, where the representation of a continuous distribution in spiking networks comes from the continuously smooth tuning curves of neurons.*
> > > >
> > > > We agree that digital-to-analogue conversion is a fundamental conceptual issue in trying to approximately sample from a continuous distribution using a spiking network, as we noted in the manuscript. Here, and in prior work on EBNs and other spiking networks, our approach to this problem is quite simple: the approximate samples are low-pass filtered spike trains. This does not eliminate DAC error, but, as we show empirically, yields a reasonable approximation for moderately large networks. We also note that estimating moments of the distribution from samples through temporal averaging introduces an additional smoothing step.
> > > >
> > > > As described in the Introduction of the submitted manuscript, probabilistic population codes afford an entirely distinct strategy for representing probability distributions using spiking neural networks. As noted there, our focus in this manuscript is on sampling-based codes because of their better potential scalability to higher dimensions, as noted in previous work (Fiser et al. 2010, Beck et al. 2011). To clarify this point, we have revised the corresponding paragraph of the Introduction (Lines 24-34), which now reads as follows:
> > > >
> > > > > Several neural architectures for probabilistic computation have been proposed, including: probabilistic population codes (Ma et al. 2006), which in certain cases allow a direct readout of uncertainty; direct encoding of metacognitive variables, such as decision confidence [...]; doubly distributional codes [...] which distinguish uncertainty from multiplicity; and sampling-based codes [...], where the variability in neural dynamics corresponds to a signature of exploration of the posterior probability. Most experiments quantifying uncertainty representations in single biological neurons have only varied parameters along one or two dimensions, such as in Bayesian cue combination [...]. In these conditions, many algorithms can perform adequately. However, probabilistic inference becomes more challenging as the entropy of the posterior distribution---which often scales with dimensionality---increases [...]. Some algorithms that work well in low dimensions, such as probabilistic population codes, may scale poorly to high-dimensional settings (Fiser et al. 2010).
> > > >
> > > > *[...]*

---

> ### Comment · Reviewer_jyfu · 2022-08-09
> **Thanks for the authors' reply**
>
> The authors' reply addressed some of my confusion, e.g., the influence of the D matrix in Eq. 14 on the sampling dynamics,  the denominator in Eq. 4, and the natural geometry has a smaller discretization error than naive sampling.

---

### Official Review · Reviewer_TEk8 · 2022-07-12

**Rating:** 6
**Confidence:** 5
**Soundness:** 3 good
**Presentation:** 3 good
**Contribution:** 3 good

**Summary:**

New framework for constructing fast sampling dynamics in spiking neural networks, encompasses many of the previously proposed solutions as limit cases but also allows for a unified treatment and interesting variations.

**Questions:**

- decoding requires low pass filtering of spikes, which means that the filter width puts a hard limit on the autocorrelation function of the outputs, same as in distributed sampling a la savin et al. Why is then this faster? is it because the time constant is not tied to the membrane time constant the way it needs to be there?

- explicit rejection step in MH seems to require fundamentally global knowledge which affects classic criteria for biological plausibility, is the discretization of time that assumes asynchronous updates making this local again?

-  hard to intuitively get the source of the speedup, especially given the earlier proofs of hennequin that langevin is the slowest possible random walk with a given gaussian stationary distribution, how does the structure of the covariance play with the  sample autocorrelation function?

- is there a way to make more formally precise statements about sampling speed in EBNs for the different variants or is this mainly relying on the ma work for generic speedup arguments?

**Limitations:**

no issues

**Strengths And Weaknesses:**

Strengths:
- clear biological and computational motivation
- interesting knowledge transfer between machine learning and computational neuroscience
- interesting mechanics: deterministic dynamics, stochastic spike generation process (usually stochastic dynamics, deterministic theshold)

Weaknesses:
- restricted to multivariate gaussian posteriors, although the time varying mean makes it somewhat more unusual/interesting as a setup
- dependence of input (the actual inference part) missing in the construction (for simplicity i would agree this is fine for static posteriors but seems awkward once there is a time constant in the posterior changes themselves)
- emphasis on the practically relevant scale of sampling being unexplored is an overstatement of fact, both the hennequin and savin type of dynamics can be used to extract moments of interest at time scale of hundreds of milliseconds (explicitly with spiking neurons at least for the second case), which many would argue is perceptually relevant enough, especially given the inherent tradeoff between precision and time for all sampling based codes; that is not to say that there is no room for alternative models of fast sampling but it's not nearly as bad as the introduction would make you believe
- very precise/artificial architectural constraints on the solution (pair of readout pools tied weights etc)
- in the neurally relevant regime the dynamics are not guaranteed to have the target posterior statistics. unclear why the series of approximatons should be expected to be constrained rather than lead to accumulation of errors and big deviations from the target

Minor:
- strictly speaking it is not the dimensionality of the latent space per se but rather posterior entropy that limits sampling speed, although admittedly the two tend to go together in simple cases
- this is semantics i would say that typically the goal of bayesian perception is stated as computing a posterior over latent variables, not parameters
- potentially relevant refs: radford neal tech report on speeding up sampling by dropping the detailed balance requirement, and the old lars buesing sampling paper on a biological realization of that idea (refractory period of sampling).

---

> ### Author Response · Authors · 2022-08-02
> **Response to Reviewer TEk8 (Part 1 of 4)**
>
> *[...]*
>
> ### Strengths And Weaknesses:
>
> #### Strengths:
>
> *[...]*
>
> We thank the referee for their careful and favorable assessment of our work.
>
> #### Weaknesses
>
> - *restricted to multivariate gaussian posteriors, although the time varying mean makes it somewhat more unusual/interesting as a setup*
>
>     We thank the reviewer for raising this important point. As the reviewer points out, the implementation with time varying mean would allow non-Gaussian posteriors to be estimated through sampling. Indeed the dynamics encoded in the network could correspond to Langevin sampling within the complete recipe framework of any probability distribution in the exponential family. We chose here to restrict ourselves to the multivariate Gaussian as they are better understood in the statistics and machine learning literature, and can be implemented in linear integrate-and-fire networks. More complex distributions will require non-linearities and therefore more complex (but still biologically plausible) neurons. We did not add a sentence to discuss the point due to the length constraint, but would be happy to do so in the final manuscript.
>
> - *dependence of input (the actual inference part) missing in the construction [...]*
>
>     In our construction, the input to the network appears through the time-varying mean ${\theta}(t)$. In the linear Gaussian model we discuss in Lines 71-80 of Section 2 and detail in Appendix C.2-3, the network infers the expected value of the parameters/latent variables given a sensory input $\textbf{x}$, of which the target mean ${\theta}$ is a linear function (see also Hennequin et al. and Savin and Denève, who use this model). In this simple setting, the covariance structure does not change with the input. The new Figure E.1 (https://figshare.com/s/15ccd407eb0ba614d23d) highlights this point. The estimate of the variance is perturbed by stimulus onset (Figure E.1b) but relaxes to its steady state value. Note that this relaxation is both faster and more accurate in the natural geometry condition. For non-Gaussian posteriors, the variance would be changing with the input as well.
>
>     Note that the relative timescale of changes in the posterior is the important point. Our result show that given a change in input, the network implementing the inference in the natural space of parameters samples the new posterior within 50 ms for a large range of parameters. As long as the changes in the posterior occur at this speed (the speed of perception), it will be adequately sampled.
>
>     If the reviewer deems it necessary, we are happy to include an extended discussion of these points in the Introduction and discussion of the final version of the manuscript; we have not done so thus far because of space constraints.
>
> - *emphasis on the practically relevant scale of sampling being unexplored is an overstatement of fact, both the hennequin and savin type of dynamics can be used to extract moments of interest at time scale of hundreds of milliseconds [...]*
>
>     We agree with the reviewer that the works of Hennequin et al. and Savin and Denève have made an important contribution towards sampling at the speed of perception in biological neural networks, but we believe our contribution goes beyond these papers in important ways. Hennequin et al. (2014) use a rate network and only assess the auto-correlation of the samples provided by the network. They do not quantify whether these samples yield good estimates of statistics of the target probability distribution. Savin and Denève (2014) also makes a great contribution by implementing parallelized Langevin samplers within the efficient balanced network (EBN) framework. However, they do not show that their approach scales to high-dimensional (or high posterior entropy) parameter spaces as they only quantify performance in two dimensions. We have slightly reworded our presentation of these papers in Lines 35-45 of the Introduction to clarify our contributions:
>
>     > Of the proposed approaches to probabilistic computation in neural networks, sampling-based codes are grounded in the strongest theoretical framework [...], and have been used to perform inference at scale [...]. Previous works have proposed several approaches to accelerate sampling in biologically-inspired algorithms. [Hennequin et al.] showed that adding non-reversible dynamics to rate networks can reduce the sample autocorrelation time. However, they did not study convergence of the sampling distribution, and did not consider the biologically-relevant setting of spiking networks. [Savin and Denève] used a distributed code to parallelize sampling in spiking networks, but only considered two-dimensional distributions. Therefore, it remains unclear how accurate sampling from high-dimensional distributions at behaviorally-relevant timescales can be achieved using spiking networks.

---

> > ### Author Response · Authors · 2022-08-02
> > **Response to Reviewer TEk8 (Part 2 of 4)**
> >
> > #### Weaknesses (Part 2 of 2)
> >
> > - *very precise/artificial architectural constraints on the solution (pair of readout pools tied weights etc)*
> >
> >     We agree that our construction assumes strong constraints on the network architecture. As noted by the referee, these constraints are useful because they allow a relatively straightforward construction. As noted in Lines 300-320 of the Discussion, more careful investigation of how asymmetry of weights affects the performance of the probabilistic spiking network will be an interesting subject for future work, particularly in the context of biologically-relevant constraints (e.g., Dale's law). We also note that the EBN-based sampler does not rely on having fine-tuned balancing of readout weights.
> >
> > - *in the neurally relevant regime the dynamics are not guaranteed to have the target posterior statistics. unclear why the series of approximations should be expected to be constrained rather than lead to accumulation of errors and big deviations from the target*
> >
> >     Thank you for this comment. We agree entirely that it is not \emph{a priori} clear that our approximations should yield constrained rather than accumulating error, and that we rely on numerical evidence to support this claim. We note that this is similar to previous work on sampling using EBNs; Savin and Denève do not provide analytical guarantees their procedure yields the desired stationary distribution. See also the point below on the difficulty of formal analysis of convergence in EBNs.
> >
> > ### Minor:
> >
> > - *strictly speaking it is not the dimensionality of the latent space per se but rather posterior entropy that limits sampling speed, although admittedly the two tend to go together in simple cases*
> >
> >     Thank you for this comment; we have updated the wording of Lines 31-34 of the Introduction to more precisely mention posterior entropy. The relevant sentence now reads as follows: "However, probabilistic inference becomes more challenging as the entropy of the posterior distribution---which often scales with dimensionality---increases."
> >
> > - *this is semantics i would say that typically the goal of bayesian perception is stated as computing a posterior over latent variables, not parameters*
> >
> >     We thank the reviewer for their comment. We have reworded the introduction in Section 2 to make this link between the terms used in the statistics and machine learning literature and those from the Bayesian perception literature more explicit; the relevant sentence now reads as follows: "The goal of a neural network performing probabilistic inference is to estimate a posterior distribution $P({\theta} | \mathbf{x})$ over $n_p$ latent variables (or parameters) ${\theta}$ given an input $\mathbf{x}$."
> >
> > - *potentially relevant refs: radford neal tech report on speeding up sampling by dropping the detailed balance requirement, and the old lars buesing sampling paper on a biological realization of that idea (refractory period of sampling).*
> >
> >     Thank you for these suggestions; we now cite both of these references in our updated manuscript (we previously cited only Buesing et al). We have added a comment referencing these works when we introduce Metropolis-Hastings sampling on Lines 98-99 of Section 2.1: "We note that previous work has shown that violations of detailed balance can accelerate sampling (Neal 2004, Buesing et al. 2011, Pecevski, Buesing, and Maass 2011), but we will not carefully explore this possibility in the present work." We also now mention in the Discussion that "Further investigation of how violations of detailed balance through these mechanisms and the matrix $\mathbf{S}$ in the 'complete recipe' framework could enable faster sampling will be a particularly interesting objective." As the referee appreciates, analysis of Markov chains that violate detailed balance remains challenging.

---

> > > ### Author Response · Authors · 2022-08-02
> > > **Response to Reviewer TEk8 (Part 3 of 4)**
> > >
> > > ### Questions
> > >
> > > - *decoding requires low pass filtering of spikes, which means that the filter width puts a hard limit on the autocorrelation function of the outputs, same as in distributed sampling a la savin et al. Why is then this faster? is it because the time constant is not tied to the membrane time constant the way it needs to be there?*
> > >
> > >     The speedup in the Savin-Denève network was due to the parallelization of the inference. By having more neurons than latent variables/parameters the network can have effectively faster sampling. However, this is fundamentally limited by the dynamics of the underlying Langevin process approximated by the network and fails in high-dimensions for naïve Langevin dynamics. By introducing the natural geometry, our network approximates a sampling process with much more favorable dynamics as shown in Figure 2, 4, and E.1. We gave a new analysis of sampling in linear rate networks in Appendix D that makes the effect of natural geometry apparent; please see our discussion of these results below.
> > >
> > > - *explicit rejection step in MH seems to require fundamentally global knowledge which affects classic criteria for biological plausibility, is the discretization of time that assumes asynchronous updates making this local again?*
> > >
> > >     Thank you for your question. Once we write the acceptance ratio as a function of membrane potentials and thresholds (equations 5-6) we only require local knowledge to compute the acceptance ratio for each neuron. The issue of instantaneous communication between neurons does remain, and is a shortcoming of EBN networks which recent work has worked to overcome (Rullán Buxó and Pillow 2020; see also our answer to your last question).
> > >
> > > - *hard to intuitively get the source of the speedup, especially given the earlier proofs of hennequin that langevin is the slowest possible random walk with a given gaussian stationary distribution, how does the structure of the covariance play with the sample autocorrelation function?*
> > >
> > >     Thank you for this question. We first recall that Hennequin et al. (2014)'s analysis is restricted to samplers with isotropic noise covariance, and strictly speaking shows only that naïve Langevin sampling lies at a stationary point of their slowness cost. To show how the structure of the covariance affects the sample autocorrelation and rate of convergence in setting of Hennequin et al. (2014), we have added a new analysis of sampling from Gaussian distributions in linear rate networks to Section 3.1, with details in a new Appendix D. This shows how large eigenvalues of the target covariance matrix introduce slow timescales in naïve Langevin dynamics, to which sampling in the natural space are insensitive.
> > >
> > >     Concretely, consider sampling from a Gaussian with mean zero and covariance matrix ${\Sigma}$. Using the naïve Langevin dynamics
> > >
> > >     $$d\mathbf{z}(t) = - {\Sigma}^{-1} \mathbf{z}\,dt + \sqrt{2} d\mathbf{W}(t),$$
> > >
> > >     the stationary autocovariance of the samples is
> > >
> > >     $$\mathbf{C}\_{s}(t-t') = \mathbb{E}\_{s} \mathbf{z}(t) \mathbf{z}(t')^{\top} =  e^{-{\Sigma}^{-1} |t-t'|} {\Sigma}$$
> > >
> > >     and the 2-Wasserstein distance between the ensemble distribution of samples at time $t$ for initial condition $\mathbf{z}(0) = \mathbf{0}$ and the target distribution is
> > >
> > >     $$W_{2}(t) = \sqrt{ \sum_{i=1}^{n_p} \sigma_{i} \left[ 1 - (1 - e^{-2 t / \sigma_{i}})^{1/2} \right]^{2} } ,$$
> > >
> > >     where $\sigma_{i}$ are the eigenvalues of ${\Sigma}$. In contrast, for a network that samples in the natural space,
> > >
> > >     $$d\mathbf{z}(t) = - \mathbf{z}\, dt + \sqrt{2 {\Sigma}} d\mathbf{W}(t),$$
> > >
> > >     the stationary covariance is
> > >
> > >     $$\mathbf{C}\_{s}(t-t') = e^{-|t-t'|} {\Sigma}$$
> > >
> > >     and the $W_{2}$ distance is
> > >
> > >     $$W_{2}(t) = \sqrt{\textrm{tr}\, {\Sigma}} [1 - (1-e^{-2t})^{1/2} ] .$$
> > >
> > >     Thus, it is easy to see that large covariance eigenvalues introduce long timescales in the stationary autocorrelation and in the ensemble $W_{2}$ distance for naïve Langevin sampling, while the timescale of sampling in the natural space is insensitive to these eigenvalues. The details of this analysis are given in Appendix D of our updated manuscript.

---

> > > > ### Author Response · Authors · 2022-08-02
> > > > **Response to Reviewer TEk8 (Part 4 of 4)**
> > > >
> > > > ### Questions (Part 2 of 2)
> > > >
> > > > - *is there a way to make more formally precise statements about sampling speed in EBNs for the different variants or is this mainly relying on the ma work for generic speedup arguments?*
> > > >
> > > >     Thank you for this question. We very much agree with the referee that it would be desirable to make precise statements about sampling speed in EBNs, but it is not entirely clear how such an analysis would proceed. We rely on the Ma et al. work for a generic argument for speedup of the sampling process approximated by the EBN (see also our new analysis of speedup in linear rate networks), but this of course does not transfer to the EBN because of the discretization inherent in spiking. Thus, as in most prior work on EBNs, we rely on numerical evidence for the claim that natural gradient yields a speedup. Recent work by Calaim, Dehmelt, Gonçalves, and Machens (eLife 2022) demonstrates a simple method to show that EBN approximation error of deterministic processes is bounded under certain constraints, but their analysis employs a different spiking rule (instead of constraining only one neuron to spike at a time, they enforce a refractory period during which a neuron is not allowed to spike). We will add a discussion paragraph to the final version of our manuscript to mention the possibility of formally analyzing sampling speed in EBNs; we are currently unable to do so due to space constraints. We note that we can provide some formal analysis of how sampling from highly correlated distributions fails in geometry-blind samplers using the probabilistic spiking rule; we argue in Appendix B.5 that the spike acceptance probability should tend to zero.
> > > >
> > > > *[...]*

---

> > > > > ### Comment · Reviewer_TEk8 · 2022-08-06
> > > > > **Post rebuttal comments**
> > > > >
> > > > > Thanks for the clarifications. I continue to like the ambition of the paper but still feel that the spelling out of the additional benefits of this work relative to recent fast sampling procedures could be sharper and avoid some semantics type overstatement (e.g. just because large simulations of high d were not included in the previously mentioned papers there was nothing fundamental preventing them from doing it, and they are formally quite similar to this solution).

---

> > > > > > ### Author Response · Authors · 2022-08-08
> > > > > > **Reply to post rebuttal comments**
> > > > > >
> > > > > > The reviewer is right that the efficient balanced network limit of our implementation has similarities to the one proposed by Savin &  Denève. However the addition of dynamics in the natural space of parameters is specifically what allows the network to perform for high-dimensional distributions. Although high-d simulation can be implemented in the naïve Langevin framework of Savin & Denève, in practice the accuracy of the sampling degrades rapidly with the entropy of the posterior distribution (either due to increased covariance or increased dimensionality as shown in Figure 2 of our manuscript) and sampling using the natural geometry is needed to get reasonable accuracy. The parallelization they introduce does accelerate the sampling as it performs inference in parallel chains but it will not improve the accuracy and our simulations show that the naïve approach fails in high-dimensions. Furthermore, the parallel approach requires $K$ times as many neurons where $K$ is the number of parallel chains.
> > > > > >
> > > > > > We propose to rewrite the paragraph introducing these previous works:
> > > > > >
> > > > > > >Previous works have proposed several approaches to accelerate sampling in biologically-inspired algorithms. For example, Hennequin et al. [9] showed that adding non-reversible dynamics to rate networks can reduce the sample autocorrelation time. Savin and Denève [20] used a distributed code to parallelize sampling in spiking networks based on efficient balanced networks, however they only considered two-dimensional distributions and while this approach accelerates sampling its accuracy will degrade in high-dimensions. Beyond this initial work, a general framework for sampling in biologically-inspired spiking neural networks incorporating recent advances from the machine learning literature remains to be developed.
> > > > > >
> > > > > > We believe our contribution is a significant advance from previous work due to both the general framework we introduce as well as the dynamics within the complete recipe framework. We hope the reworded paragraph should focus the reader on our key contributions as highlighted in lines 53-62:
> > > > > > - The general framework linking sampling in Metropolis-Hastings based networks and Efficient Balanced networks
> > > > > > - Introducing the importance of the population geometry in controlling the speed and accuracy of convergence.
> > > > > >
> > > > > > We were also wondering if the reviewer had comments on the presentation of our new derivations providing intuition for the speedup in the natural geometry setting.

---

### Official Review · Reviewer_Pjbs · 2022-07-15

**Rating:** 6
**Confidence:** 1
**Soundness:** 3 good
**Presentation:** 2 fair
**Contribution:** 2 fair

**Summary:**

In this paper, the authors:
- construct a spiking neural network that performs Metropolis-Hastings sampling of a target posterior distribution (for, e.g., estimating external features given noisy sensory input), and extend it to continuous time dynamics
- demonstrate its relationship to efficient balanced networks
- show how the network can be augmented (by imposing population geometry) to implement the "generalized" stochastic gradient MCMC
- and demonstrate that geometry-aware networks are superior to geometry-naive networks, especially with increasing correlation and posterior dimension


**Questions:**

I'm somewhat confused about the central point of the paper, that population geometry improves sampling efficiency. If I understand correctly, population geometry, in the most important sense, means the way in which the sampling space can be projected (e.g., to the principal components, or some other non-Euclidean manifold) to essentially decorrelate the dimensions of the posterior (in the case of the correlated Gaussian).

If this is correct, then I would suggest for the authors to more clearly and explicitly mention this much earlier in the paper (apologies if I I had missed it). Figure 1C is helpful in this regard, but I'm still uncertain if I really got what "population geometry" means, since typically, in the neuroscience context, population geometry refers to a e.g., a lower dimensional manifold on which population activity resides, and I'm not sure if this idea is invoked at all in this paper.

Furthermore, and this is probably due to my naivety still, I'm a bit confused as to how the network should have access to the natural geometry of the problem (in the form of the projection B, or D)? Similarly, it's a bit unclear how the weights of the network should be learned, and whether it is dependent on the problem. From the perspective of a naive reader unfamiliar with this subfield of comp neuro, it would be very helpful if the authors took a more descriptive approach to provide intuition for such basic high-level ideas.

Overall, I believe my lack of expertise hinders my ability to understand this paper's contribution. However, I think I am not alone in that, unless one works exactly in the field of spiking network samplers, it is a bit difficult to understand the paper even at a high level.

**Limitations:**

Extensive discussion of the paper's limitations and assumptions, and no discussion of negative societal impact (though I don't foresee any immediate impact).

**Strengths And Weaknesses:**

I'm quite naive to the field of spiking network as samplers, so I cannot really evaluate the originality, significance, or the technical quality of the work. Broadly speaking, I believe the work is of interest to the computational neuroscience community, since "the Bayesian brain" is an active field of research, and therefore, a central question is how a biologically realistic network (i.e., spiking network) could implement such sampling dynamics. In addition, at a superficial level, the work appears to be carefully done and cites relevant literature, as well as carefully discussing its relations to previous work, and its limitations. Based on the figures, it seems that the central claim of the paper is supported, that access to the natural geometry of the sampling space enables more efficient and accurate sampling.

There are several assumptions that seem biologically implausible to me, many of which the authors touch on in the discussion, and I understand certain assumptions are required for such theoretical work and cannot comment on whether they are standard for the field. However, on the issue of long membrane time constant (nu<<1), it would be nice to see if/when the results break down, with a shorter integration window. Similarly, the fact that only one neuron is allowed to spike at a time significantly dampens my enthusiasm for the work, since at some point the actual model become so far away from the (very nicely stated) high level motivation.

There are some low-level editing mistakes that can improve the quality of the work, e.g., Figure 4 panels a and b are labeled as b and c.

---

> ### Author Response · Authors · 2022-08-02
> **Response to Reviewer Pjbs (Part 1 of 2)**
>
> *[...]*
>
> ### Strengths And Weaknesses
>
> *I'm quite naive to the field of spiking network as samplers, so I cannot really evaluate the originality, significance, or the technical quality of the work. Broadly speaking, I believe the work is of interest to the computational neuroscience community, since "the Bayesian brain" is an active field of research, and therefore, a central question is how a biologically realistic network (i.e., spiking network) could implement such sampling dynamics. In addition, at a superficial level, the work appears to be carefully done and cites relevant literature, as well as carefully discussing its relations to previous work, and its limitations. Based on the figures, it seems that the central claim of the paper is supported, that access to the natural geometry of the sampling space enables more efficient and accurate sampling.*
>
> We thank the reviewer for carefully reading our paper and for their comments. We hope we address their concerns below.
>
> *There are several assumptions that seem biologically implausible to me, many of which the authors touch on in the discussion, and I understand certain assumptions are required for such theoretical work and cannot comment on whether they are standard for the field. However, on the issue of long membrane time constant ($\eta \ll 1$), it would be nice to see if/when the results break down, with a shorter integration window. Similarly, the fact that only one neuron is allowed to spike at a time significantly dampens my enthusiasm for the work, since at some point the actual model become so far away from the (very nicely stated) high level motivation.*
>
> Thank you for these comments. With regards to the membrane time constant, we note that we are assuming that the decay constant in discrete time $\eta$ is small. In terms of the membrane time constant $\tau_{m}$ and timestep between spike proposals $\Delta$, we have $\eta = \Delta/\tau_{m}$, hence what we assume is that the membrane time constant is long relative to the interval between updates, not that it is long in an absolute sense. Indeed, in Section 2.2, we take the continuum limit $\Delta \downarrow 0$ in which spike proposals are made infinitely often. Therefore, our model does not require membrane time constants $\tau_{m}$ that are longer than is plausible biologically.
>
> The fact that only one neuron is allowed to spike at a time is not a severe constraint relative to prior works, as it is standard in studies of efficient balanced networks (EBNs). As mentioned in Lines 280-288 of the Discussion, the constraint that only one neuron is allowed to spike at each timestep is present in previous work on efficient balanced networks (Boerlin et al 2013, Savin and Denève 2014), and some mechanism for constraining firing rate is present in all works on the subject (Rullán Buxó and Pillow 2021, Calaim et al 2022). As noted there, relaxing these constraints will be an important objective for future work.
>
> *There are some low-level editing mistakes that can improve the quality of the work, e.g., Figure 4 panels a and b are labeled as b and c.*
>
> Thank you for noticing this typo, which has been fixed. We have edited the manuscript for correctness throughout.

---

> > ### Author Response · Authors · 2022-08-02
> > **Response to Reviewer Pjbs (Part 2 of 2)**
> >
> > ### Questions
> >
> > *I'm somewhat confused about the central point of the paper, that population geometry improves sampling efficiency. If I understand correctly, population geometry, in the most important sense, means the way in which the sampling space can be projected (e.g., to the principal components, or some other non-Euclidean manifold) to essentially decorrelate the dimensions of the posterior (in the case of the correlated Gaussian).*
> >
> > *If this is correct, then I would suggest for the authors to more clearly and explicitly mention this much earlier in the paper (apologies if I I had missed it). Figure 1C is helpful in this regard, but I'm still uncertain if I really got what "population geometry" means, since typically, in the neuroscience context, population geometry refers to a e.g., a lower dimensional manifold on which population activity resides, and I'm not sure if this idea is invoked at all in this paper.*
> >
> > We thank the reviewer for their comment. It is indeed the geometry of the "lower dimensional manifold on which population activity resides" that will govern the convergence dynamics of the sampling process. To clarify this point, we propose to change the title of our paper to "Natural gradient enables fast sampling in spiking neural networks." Moreover, we have reworded the paragraph in the Introduction discussing population geometry (Lines 46-52). We highlight that the geometry imposed by the population should ideally correspond to the geometry of the natural space of parameters as defined in the information geometry literature:
> >
> > > In this paper, we show how the choice of the geometry of neural representations at the population level [36, 37], set by the neural code and the neural dynamics, can accelerate sampling-based inference in spiking neural networks. Ideas from information geometry allow us to perform inference in the natural space of parameters, which is a manifold with distances measured by the Fisher-Rao metric (Figure 1.c) [38–41]. Concretely, we leverage recently-proposed methods for accelerating sampling from the machine learning literature [35,41–43] to design novel efficient samplers in spiking neural networks.
> >
> >
> > *Furthermore, and this is probably due to my naivety still, I'm a bit confused as to how the network should have access to the natural geometry of the problem (in the form of the projection B, or D)? Similarly, it's a bit unclear how the weights of the network should be learned, and whether it is dependent on the problem. From the perspective of a naive reader unfamiliar with this subfield of comp neuro, it would be very helpful if the authors took a more descriptive approach to provide intuition for such basic high-level ideas.*
> >
> > Thank you for this question. In this work we focus on inference, given prior expectations and sensory input, of the probabilities of the values of the parameters/latent variables according to the Gaussian linear model presented in Appendix C. Learning the statistics of the world model (priors and affinity matrix $\mathbf{A}$) can be done by learning at a slower timescale, as we mention in Lines 321-331 of the Discussion. We remark that this could extend to evolutionary timescales. Note that the expression of the complete recipe decouples the connectivity due to the gradient of the energy function ($\nabla U_{\Theta}$) and the matrices controlling the geometry ($\mathbf{D}$ and $\mathbf{B}$). As shown in the work by Gong et al. (ICLR, 2019) that we cite in the discussion, these parameters could be learned independently via meta-learning. Expanding these approaches for spiking neural networks would be an interesting avenue for future work. Space permitting, we would be happy to extend our discussion of these issues in the final version of the submission.
> >
> >
> > *Overall, I believe my lack of expertise hinders my ability to understand this paper's contribution. However, I think I am not alone in that, unless one works exactly in the field of spiking network samplers, it is a bit difficult to understand the paper even at a high level.*
> >
> > Thank you for this comment. We hope that our revisions have enhanced the readability of the manuscript, and appreciate your suggestions for how it might be further improved.
> >
> > ### Limitations
> >
> > *Extensive discussion of the paper's limitations and assumptions, and no discussion of negative societal impact (though I don't foresee any immediate impact).*
> >
> > Thank you for your comments. As we noted in the Checklist, "Our work is purely theoretical, and we do not anticipate that it will have negative societal impacts as outlined in the ethics guidelines." We would be happy to include a sentence to that effect in the finalized version of the manuscript.
> >
> > *[...]*

---

> > > ### Comment · Reviewer_Pjbs · 2022-08-08
> > > **thanks for the response**
> > >
> > > I appreciate the authors addressing my concerns and suggestions in earnest. The response and changes have improved the high-level clarity of the paper, and helped me understand the context of the existing literature, and the authors' contributions. I still have doubts about some assumptions and how they can ultimately be realized in the biological brain, but given the references the authors have provided, I assume these are standard for the field, and should not negatively impact ratings of the current work. I have updated my scores accordingly, but remain unconfident in my assessment and defer to the other reviewers & AC.

---

> > > > ### Author Response · Authors · 2022-08-08
> > > > **Thank you for the comments**
> > > >
> > > > We thank the reviewer for helping us improve the clarity and presentation of our paper.

---

### Author Response · Authors · 2022-08-02
**Response to common concerns for "Population geometry enables fast sampling in spiking neural networks"**

We thank the referees for their thoughtful reviews of our paper. We have uploaded a revised manuscript that addresses their concerns and strengthens our opinion that this paper should be presented at NeurIPS. This comment describes major additions to our manuscript; we reply in detail to specific referee comments individually.

- In response to **Reviewer Pjbs**' questions regarding the meaning of `population geometry' we propose to change the title of our manuscript to ``Natural gradient enables fast sampling in spiking neural networks.'' We think this modification will help resolve possible confusion, and appreciate the referees' feedback on this point.

- The most substantial addition to the content of our manuscript is a new analysis of how natural gradient enables fast sampling in linear rate networks. This addresses a question raised by **Reviewer TEk8** regarding an intuition about the source of the speedup in sampling, with particular reference to the prior work of Hennequin et al. (NeurIPS 2014). We introduce these results at the end of a reworked section 3.1 and provide a detailed derivation in Appendix D (Note that the numerical details and supplementary figures are now in Appendix E).

- We have added a supplemental figure (Figure E.1; see anonymous upload at https://figshare.com/s/15ccd407eb0ba614d23d) showing the timestep-by-timestep timecourses of distribution estimates for the EBN sampler. This figure allows the reader to better put into context the parameter sweeps of the distribution estimates at $t=50$ ms and $t=1.5$ s after stimulus onset shown in Figures 2 and E.3

- We have added another supplemental figure (Figure B.1) showing how the scale of the readout matrix affects the probabilistic spiking rule. This figure provides further characterization of the behavior of the novel Metropolis-Hastings based sampler we propose.

- We added a number of clarifications as highlighted in the responses to individual reviewers. Note that to improve readability, we now denote samples by $z_t$, rather than $\hat{\theta}_t$ as in our submitted manuscript, to avoid confusion with the target $\theta_t$.

---

### Meta-Review · Area_Chair_vJPn · 2022-08-26

**Recommendation:** Accept
**Confidence:** Less certain

**Metareview:**

Although some reviewers have reservations about strong modelling assumptions, the main contribution of the paper is clearly presented and technically sound.

**Award:**

No

---

### Decision · Program_Chairs · 2022-09-14

Accept